# Trade-Off Between Entropy and Gini Index in Income Distribution

**DOI:** 10.3390/e28010035

**Published:** 2025-12-26

**Authors:** Demetris Koutsoyiannis, G.-Fivos Sargentis

**Affiliations:** Department of Water Resources and Environmental Engineering, School of Civil Engineering, National Technical University of Athens, Heroon Polytechneiou 5, 15772 Zographou, Greece; fivos@itia.ntua.gr

**Keywords:** entropy, principle of maximum entropy, K-moments, stochastics, wealth, income profiles, Gini index, inequality, stability

## Abstract

We investigate the fundamental trade-off between entropy and the Gini index within income distributions, employing a stochastic framework to expose deficiencies in conventional inequality metrics. Anchored in the principle of maximum entropy (ME), we position entropy as a key marker of societal robustness, while the Gini index, identical to the (second-order) K-spread coefficient, captures spread but neglects dynamics in distribution tails. We recommend supplanting Lorenz profiles with simpler graphs such as the odds and probability density functions, and a core set of numerical indicators (K-spread $K_{2}/\mu$, standardized entropy $\Phi_{\mu }$, and upper and lower tail indices, ξ, ζ) for deeper diagnostics. This approach fuses ME into disparity evaluation, highlighting a path to harmonize fairness with structural endurance. Drawing from percentile records in the World Income Inequality Database from 1947 to 2023, we fit flexible models (Pareto–Burr–Feller, Dagum) and extract K-moments and tail indices. The results unveil a concave frontier: moderate Gini reductions have little effect on entropy, but aggressive equalization incurs steep stability costs. Country-level analyses (Argentina, Brazil, South Africa, Bulgaria) link entropy declines to political ruptures, positioning low entropy as a precursor to instability. On the other hand, analyses based on the core set of indicators for present-day geopolitical powers show that they are positioned in a high stability area.

*κατὰ μὲν τὴν οὐσίαν καὶ τὸν λόγον τὸν τὸ τί ἦν εἶναι λέγοντα μεσότης ἐστὶν ἡ ἀρετή, κατὰ δὲ τὸ ἄριστον καὶ τὸ εὖ ἀκρότης.* (*In terms of its essence and the definition of its nature, virtue is the mean, but in terms of excellence and rightness,* [virtue is] *the extreme*)Aristotle [1]

## 1. Introduction

The modern world is characterized by high interdependent complexities upon which its social prosperity is founded [2]. In this context, events of varying scales can trigger cascading effects, leading to dynamics of collapse that impact the entire social structure [3,4]. One of the most critical and intricate parameters, which has been the subject of extensive theoretical and empirical analysis, is income distribution, a foundational factor for social stability [5,6,7].

Based on income distribution, we can derive useful indices describing in an intuitive manner important aspects of a country’s society such as (in)equality and (in)stability. Among these indices, of particular interest are entropy and the Gini coefficient. Our previous research (Koutsoyiannis and Sargentis [7]) delineated entropy’s role across physical and economic systems. Extending this, here we investigate the dynamics between entropy and the Gini coefficient across income structures.

The Gini index, here reformulated as the second-order K-moment standardized by the mean, $K_{2}/\mu$ (see Section 3.1.3), and referred to as the K-spread coefficient, leads inequality assessments. Still, it masks extremes, i.e., the tail behaviour of the distributions: profiles with matching Gini indices can show stark contrasts in income extremes. Entropy instead gauges overall variability and uncertainty and, through the principle of maximum entropy (ME), identifies the likeliest—and, hence, most resilient—structure under given constraints. The ME principle was originally formalized by Jaynes [8] as a method to infer the most probable distribution given constraints. In economic contexts, ME posits that income distributions tend toward states of maximum entropy under real-world constraints.

Under constraints of specified mean, μ, and K-spread (Gini), $K_{2}/\mu$, we demonstrate here that entropy maximization results in a generalized half logistic (GHL) distribution, a limiting case of which is the exponential distribution. The latter materializes the peak entropy pole, as ($K_{2}/\mu =1/2,\Phi_{\mu }=1)$. The distance from this pole is another indicator of the resilience or stability of an economy, with a small distance denoting small instability.

The key conjecture in this study is that income profiles would naturally tend toward peak entropy given real constraints. On the other hand, pushing the economic state to depart from the peak entropy and creating large and persistent deviations from it herald fragility. Hence, a scale-invariant entropy form, or standardized entropy, $\Phi_{\mu }\leq 1$ [7], functions as a resilience gauge, complementing $K_{2}/\mu$ as a spread gauge. In simpler words, economic structures with low entropy will tend to higher entropy and hence are unstable, while those leading to maximum (standardized) entropy will be stable as the entropy cannot be increased further.

To ratify the key conjecture, we use data (percentile records) from the World Income Inequality Database (WIID), to which we fit flexible models (Pareto–Burr–Feller, Dagum) and extract K-moments and tail indices. We conduct country-level analyses (Argentina, Brazil, South Africa, Bulgaria) to investigate whether low entropy states evolve to higher entropy over the course of a country’s history, and whether this evolution can be linked to political ruptures. In addition we analyse the core set of indicators in countries with the largest populations (exceeding 50 million), including present-day geopolitical powers (China, India, USA, Russia, EU), to assess their stability based on the criteria developed.

The data sources are outlined in Section 2. Section 3 summarizes stochastic tools and links entropy and Gini. Section 4 details applications and stories. Section 5 weighs meanings and suggests paths ahead.

## 2. Data

The real-world applications in this study are based on the World Income Inequality Database [9,10], developed and maintained by the United Nations University World Institute for Development Economics Research. More specifically, the product used is the WIID Companion Country Dataset, which reports inequality data by country. This was selected because of its unparalleled comprehensiveness and global scope in providing income inequality statistics. The dataset encompasses 202 nations (including 4 historical entities), starting in 1947 (for USA; later for other countries) and extending to 2023, offering 2546 distinct country-year records. This broad coverage enables detailed analyses of inequality trends across diverse geopolitical contexts and historical eras, which is essential for our examination of the trade-off between entropy and the Gini index. WIID allows us to explore extreme Gini values and long-term patterns in countries like Argentina, Bulgaria, Brazil, and South Africa, as well as in major geopolitical powers such as the USA, China, EU, Russia, and India, ensuring a robust empirical foundation for applying theoretical stochastic tools to real-world income disparities.

A key reason for choosing WIID is its provision of percentile-based income data, which aligns perfectly with the methodological requirements of our research. The specific data used are those named “p1”–“p100”, which represent the income per capita (based on GDP) per percentile, standardized to have an average of 1. These facilitate the calculation of Lorenz curves, Gini indices, and entropy indicators directly. This granularity in percentiles supports our use of K-moments, the ME principle, and tail index estimations, allowing for precise comparisons between relevant probabilistic distributions. Unlike more aggregated datasets, WIID’s percentile-level detail enables us to handle grouped data effectively, reliably estimate empirical statistics, and visualize trade-offs in figures. We note though that for some countries percentile-level data are missing; these were excluded from our analyses.

WIID’s reliability and methodological rigour enable comparability across countries and over time, as well as the extraction of information for multi-country entities. Curated by inequality experts, the database incorporates adjustments for data quality and consistency, with transparent documentation in user guides, technical notes, and replication tools. This minimizes biases in cross-national comparisons, which is critical for our analysis of social stability and political histories in case studies. Widely recognized in academic research for monitoring global inequality trends, WIID outperforms alternative sources by offering both raw and processed data, ensuring that our findings on the limitations of the Gini index and the advantages of entropy-based approaches are grounded in high-quality, verifiable evidence.

## 3. Methods

### 3.1. Basic Stochastic Tools

#### 3.1.1. Distribution Function and Relative Concepts; Expectation and Moments

Let $\underline{x}$ be a stochastic (random) variable of continuous type (i.e., taking on values that are real numbers); notice that we underline stochastic variables to distinguish them from regular variables. We denote its *distribution function* (i.e., probability of non-exceedance) and its *tail function* (i.e., probability of exceedance), respectively, as:(1)$$F\left(x\right):=P\left\{\underline{x}\leq x\right\},\;\bar{F}\left(x\right):=1-F\left(x\right)=P\left\{\underline{x}>x\right\}$$
where *P* denotes probability. A useful derived function is the so-called *odds function*:(2)$$O\left(x\right):=\frac{F\left(x\right)}{\bar{F}\left(x\right)}=\frac{F\left(x\right)}{1-F\left(x\right)}$$

Both $F\left(x\right)$ and $O\left(x\right)$ are nondecreasing functions, and since the variable $\underline{x}$ is continuous, the inverse functions exist. The inverse of $F\left(x\right)$, denoted as x(F) is called the *quantile function*. The derivative of the distribution function:(3)$$f\left(x\right):=\frac{dF\left(x\right)}{dx}$$
is the *probability density function* and obeys the obvious relationship:(4)$$\int_{-\infty }^{\infty }f\left(x\right)dx=1$$

Any deterministic function of $\underline{x}$, $g\left(\underline{x}\right)$, is a stochastic variable per se, because its argument is stochastic. The *expectation* of the stochastic variable $g\left(\underline{x}\right)$ is defined as:(5)$$E\left[g\left(\underline{x}\right)\right]:=\int_{-\infty }^{\infty }g\left(x\right)f\left(x\right)dx$$For $g\left(\underline{x}\right)=\underline{x}$ and $g\left(\underline{x}\right)={\left(\underline{x}-\mu \right)}^{2}$ we get, respectively, the *mean*, μ, and the *variance*, γ, of $\underline{x}$:(6)$$\mu :=E\left[\underline{x}\right]=\int_{-\infty }^{\infty }xf\left(x\right)dx,\;\gamma :=E\left[{\left(\underline{x}-\mu \right)}^{2}\right]=\int_{-\infty }^{\infty }{\left(x-\mu \right)}^{2}f\left(x\right)dx$$The variance is necessarily nonnegative and its square root, $\sigma :=\sqrt{\gamma }$, is the standard deviation. For nonnegative variables, the limit −∞ in the above integrals is replaced by 0, while the ratio σ/μ, termed the *coefficient of variation*, is a useful dimensionless index of the variability of a system.

#### 3.1.2. Entropy and Standardized Entropy

It is possible to define a function g() in terms of not the variable $\underline{x}$ but the probability density per se, i.e., $g\left(\underline{x}\right)=h\left(f\left(\underline{x}\right)\right)$, where h() is any specified function. Among the several choices of h(), most useful is the logarithmic function, which results in the definition of *entropy*, $\Phi \left[\underline{x}\right]$. The emergence of the logarithm in the definition of entropy follows some postulates set up by Shannon (1948, [11]) for stochastic variables of discrete type. Extension for a continuous stochastic variable $\underline{x}$ was not contained in Shannon’s original work but was given later (see e.g., [12,13] p. 375) as:(7)$$\Phi \left[\underline{x}\right]:=E\left[-\ln\frac{f\left(\underline{x}\right)}{\beta \left(\underline{x}\right)}\right]=-\int_{-\infty }^{\infty }\ln\frac{f(x)}{\beta \left(x\right)}f\left(x\right)dx$$
where β(x) is a background measure density that can be any probability density, proper (with integral equal to 1, as in Equation (4)) or improper (meaning that its integral diverges). Typically, it is an (improper) Lebesgue density, i.e., a constant. We note that most texts do not include the background measure density $\beta \left(\underline{x}\right)$ in the definition (or set $\beta \left(\underline{x}\right)\equiv 1$) but, in terms of physical consistency, this is an error, because, in order to take the logarithm of a quantity, this quantity must be dimensionless. The density function has units $\left[f\left(x\right)\right]=[x^{-1}]$ and therefore we need to divide it by a quantity with the same units before taking the logarithm. Even if we choose the Lebesgue measure as the background, with $\beta \left(x\right)=1/\lambda ,$ (constant), where λ is the unit used to measure x, still the entropy depends on the unit. It can easily be verified that, if we measure x with two different units $\lambda_{1}$ and $\lambda_{2}$, the respective entropies $\Phi_{1}\left[\underline{x}\right]$ and $\Phi_{2}\left[\underline{x}\right]$ will differ by a constant:(8)$$\Phi_{1}\left[\underline{x}\right]-\Phi_{2}\left[\underline{x}\right]=\ln\frac{\lambda_{2}}{\lambda_{1}}$$

The entropy $\Phi \left[\underline{x}\right]$ per se is always dimensionless and, for continuous variables, it can be either positive or negative, depending on the assumed β(x), ranging from −∞ to a maximum value, depending on the system and, in particular, on its constraints.

Entropy quantifies uncertainty and its importance lies in the *principle of maximum entropy*, formally introduced in 1957 by Jaynes [8]. This postulates that the entropy of a stochastic system should be at maximum, under some conditions, formulated as constraints, which incorporate the information that is given about this system. The meaning of the principle is that the maximum entropy state of a system is the most probable one that is allowed by its degrees of freedom and not disallowed by its constraints. Therefore, entropy is also an index of stability: a state that is far apart from the maximum entropy state is unstable as it will tend to change toward maximizing entropy. The principle can be used for logical inference as well as for modelling physical systems. In this respect, the tendency of entropy to become maximal (as in the second law of thermodynamics), which drives natural change, can result from this principle. On the other hand, the principle equips the entropy concept with a powerful tool for logical inference.

In application in economics [7,14], for a constant background density equal to the inverse of the monetary unit (i.e., 1/λ with λ equal, e.g., to USD 1), the entropy provides an indicator of society’s wealth (even if x expresses income). If we set the background measure density to the value 1/μ, where μ is the mean income, we get the *standardized entropy*, which from Equation (8) is obtained as:(9)$$\Phi_{\mu }\left[\underline{x}\right]=\Phi \left[\underline{x}\right]-\ln\frac{\mu }{\lambda }$$This quantity, which cannot exceed a maximum value of 1 for nonnegative continuous variables (see Section 3.3), has been originally introduced [7,14] as an index of inequality. However, as we will see below, it can better be thought of as an indicator of stability, while the notion of K-moments (see next) can better characterize inequality.

#### 3.1.3. K-Moments

While in classical statistics moments of orders higher than 2 are defined and used (by substituting ${\left(\underline{x}-\mu \right)}^{p},\;p>2$, for ${\left(\underline{x}-\mu \right)}^{2}$ in Equation (6)), these cannot be reliably estimated from samples [14,15]. However, the concept of knowable moments or K-moments [15] can reliably provide estimates for high-order moments.

The K-moments are defined as follows. We consider a sample of a stochastic variable $\underline{x}$, i.e., a number p of independent copies of the stochastic variable $\underline{x}$, i.e., ${\underline{x}}_{1},{\underline{x}}_{2},\ldots ,{\underline{x}}_{p}$. If we arrange the variables in ascending order, the *i*th smallest, denoted as ${\underline{x}}_{\left(i:p\right)},\;i=1,\ldots ,p$ is termed the *i*th *order statistic*. The largest (*p*th) order statistic is:(10)$${\underline{x}}_{\left(p\right)}:={\underline{x}}_{\left(p:p\right)}=\max\left({\underline{x}}_{1},{\underline{x}}_{2},\ldots ,{\underline{x}}_{p}\right)$$
and the smallest (first) is:(11)$${\underline{x}}_{\left(1:p\right)}=\min\left({\underline{x}}_{1},{\underline{x}}_{2},\ldots ,{\underline{x}}_{p}\right)$$

We define the *upper knowable moment (K-moment) of order p* as the expectation of the largest of the *p* variables ${\underline{x}}_{\left(p\right)}$:(12)$$K_{p}^{\prime }:=E[{\underline{x}}_{\left(p\right)}]=E\left[\max\left({\underline{x}}_{1},{\underline{x}}_{2},\ldots ,{\underline{x}}_{p}\right)\right]$$
and the *lower knowable moment (K-moment) of order p* as the expectation of the smallest of the *p* variables ${\underline{x}}_{\left(1:p\right)}$:(13)$${\bar{K}}_{p}^{\prime }:=E\left[{\underline{x}}_{\left(1:p\right)}\right]=E\left[\min\left({\underline{x}}_{1},{\underline{x}}_{2},\ldots ,{\underline{x}}_{p}\right)\right]$$

An important property, directly resulting from their definition, is that the K-moments are ordered as follows:(14)$${\bar{K}}_{p}^{\prime }\leq \ldots \leq {\bar{K}}_{2}^{\prime }\leq {\bar{K}}_{1}^{\prime }=K_{1}^{\prime }=\mu \leq K_{2}^{\prime }\leq \ldots \leq K_{p}^{\prime }$$These moments are noncentral and we can also define central moments as:(15)$$K_{p}:=K_{p}^{\prime }-K_{1}^{\prime },\;{\bar{K}}_{p}:={\bar{K}}_{1}^{\prime }-{\bar{K}}_{p}^{\prime },\;K_{p},{\bar{K}}_{p}\geq 0$$

As shown in Chapter 6 in [15], for a stochastic variable $\underline{x}$ of continuous type, the upper K-moment of order *p* of $\underline{x}$, is theoretically calculated as:(16)$$K_{p}^{\prime }=pE\left[{\left(F\left(\underline{x}\right)\right)}^{p-1}\underline{x}\right]=p\int_{-\infty }^{\infty }{\left(F\left(x\right)\right)}^{p-1}x\;f(x)dx=p\int_{0}^{1}x(F)F^{p-1}dF$$Likewise, the lower K-moment of order *p* is theoretically calculated as:(17)$${\bar{K}}_{p}^{\prime }=pE\left[{\left(\bar{F}\left(\underline{x}\right)\right)}^{p-1}\underline{x}\right]=p\int_{-\infty }^{\infty }{\left(\bar{F}\left(x\right)\right)}^{p-1}x\;f(x)dx=p\int_{0}^{1}\bar{x}(\bar{F}){\bar{F}}^{p-1}d\bar{F}$$

The unbiased estimator of the upper K-moment ${\underline{K}}_{p}^{\prime }$ from a sample of size n is:(18)$${\underline{\hat{K}}}_{p}^{\prime }=\sum_{i=1}^{n}b_{inp}{\underline{x}}_{\left(i:n\right)}$$
and that of the lower K-moment is:(19)$${\hat{\bar{\underline{K}}}}_{p}^{\prime }=\sum_{i=1}^{n}b_{inp}{\underline{x}}_{\left(n-i+1:n\right)}=\sum_{i=1}^{n}b_{n-i+1,n,p}{\underline{x}}_{\left(i:n\right)}$$
where(20)$$b_{inp}=\left\{\begin{array}{ll} 0, & i<p\\ p\frac{\Gamma \left(n-p+1\right)}{\Gamma \left(n+1\right)}\frac{\Gamma \left(i\right)}{\Gamma \left(i-p+1\right)}, & i\geq p\geq 0 \end{array}\right.$$
and Γ( ) is the gamma function. For data that are grouped in classes, the resulting modified estimator is shown in Section A.1.

Based on the K-moments, we define the *K-centre of order*
p, $C_{p}$, and the *K-spread of order* p, $D_{p}$, as:(21)$$C_{p}:=\frac{K_{p}^{\prime }+{\bar{K}}_{p}^{\prime }}{2},\;D_{p}:=\frac{K_{p}^{\prime }-{\bar{K}}_{p}^{\prime }}{2}$$
where $D_{p}\geq 0$. The least-order meaningful values thereof are:(22)$$C_{1}=K_{1}^{\prime }={\bar{K}}_{1}^{\prime }=\mu ,\;D_{2}=\frac{{K_{2}^{\prime }-\bar{K}}_{2}^{\prime }}{2}=K_{2}={\bar{K}}_{2}$$Since ${\bar{K}}_{2}^{\prime }={2K}_{1}^{\prime }-K_{2}^{\prime }$ [15], we have $C_{2}=\left(1/2\right)\;\left(K_{2}^{\prime }+{\bar{K}}_{2}^{\prime }\right)=K_{1}^{\prime }=C_{1}$, i.e., the first and second order K-centre parameters are equal to each other and equal to the mean. The standardized parameter(23)$$\frac{D_{2}}{\mu }=\frac{K_{2}}{\mu }=\frac{{\bar{K}}_{2}}{\mu }$$
is a characteristic spread index, similar to the coefficient of variation σ/μ used in classical statistics, and will be referred to as the *K-spread coefficient*. Furthermore, the standardized parameter $D_{p}/D_{2}$ is also a spread index which will be referred to as the *K-spread ratio of order*
p.

#### 3.1.4. Specific Distribution Functions and Tail Indices

Here we use several distribution functions resulting from entropy maximization, which are summarized in Table 1, along with their characteristics. Among them, the three-parameter distributions, namely the Pareto–Burr–Feller (PBF) and the Dagum distributions are quite flexible and can describe most real-world systems. The Pareto, Weibull, and exponential distributions are special cases of the PBF distribution. The logistic and the generalized half logistic (GHL) constitute another form of distribution, resulting from entropy maximization with constrained mean and K-moment of order 2. The log-logistic distribution is a special case of both the PBF and Dagum distributions.

In all distributions listed in Table 1, λ is a scale parameter with dimensions identical to those of the variable $\underline{x}$, and ξ and ζ are dimensionless parameters, representing the *upper* and *lower tail indices*, respectively. For a variable $\underline{x}$ with domain (0,∞), their definitions are based on the limiting relationships:(24)$$\underset{x\to \infty }{\lim}x^{1/\xi }\bar{F}\left(x\right)=l_{U},\;\underset{x\to 0}{\lim}x^{-\zeta }F\left(x\right)=l_{L}$$
where $l_{U}$ and $l_{L}$ are nonzero and finite constants. Both can be also determined from the odds function by:(25)$$\xi =1/O^{\#}\left(\infty \right),\;\zeta =O^{\#}\left(0\right)$$
where $O^{\#}\left(x\right)$ denotes the log-log derivative (LLD) of the odds function, defined as:(26)$$O^{\#}\left(x\right):=\frac{d\left(\ln O\left(x\right)\right)}{d(\ln x)}=\frac{x\;O^{\prime }\left(x\right)}{O\left(x\right)}$$The tail indices are important characteristics of a distribution. A distribution with upper tail index ξ=0 (e.g., the exponential) is light-tailed, while one with 0<ξ<1 is a heavy-tailed distribution. In a distribution with ζ<1, the density $f\left(x\right)$ is necessarily a decreasing function, at least close to the origin, with $\underset{x\to 0}{\lim}f\left(x\right)=\infty$. In contrast, when ζ>1, the density $f\left(x\right)$ is an increasing function close to the origin, with $f\left(0\right)=0$, and is usually bell-shaped. The particular case ζ=1 is characteristic of the exponential and Pareto distributions, where $f\left(0\right)$ is finite and the density $f\left(x\right)$ is a decreasing function.

### 3.2. The Lorenz Curve and the Gini Index

The economics literature makes extensive use of the Lorenz curve and the Gini index. If x(F) is the quantile function of a probability distribution, then the Lorenz curve is simply its integral standardized by the mean, i.e.,(27)$$L\left(F\right)=\frac{1}{\mu }\int_{0}^{F}x(u)du\Leftrightarrow x\left(F\right)=\mu \;L^{\prime }(F)$$

The Gini index is the ratio of the area between the equality line and the Lorenz curve to the area under the equality line (which is 1/2), i.e.,(28)$$G=\int_{0}^{1}(F-L\left(F\right))dF/\frac{1}{2}=1-2\int_{0}^{1}L\left(F\right)dF$$
where, to obtain the rightmost result, we observe that $\int_{0}^{1}FdF=1/2$. It is easily shown (see Section A.2) that the Gini index is simply the K-spread coefficient:(29)$$G=\frac{K_{2}}{\mu }$$

Once we know the K-spread coefficient and the tail indices of a distribution, we can effectively approximate it by the following proposed relationship:(30)$$L\left(F\right)=AF\left(1-\left(1-\frac{1}{A}\right)F^{1/\zeta }\right)-\frac{A\left(1-F\right)\left({\left(1-F\right)}^{-\xi }-1\right)}{\xi }$$
where(31)$$A=\frac{\left(2-\xi \right)}{2\zeta +\xi -1}\left(\left(2\zeta +1\right)\frac{K_{2}}{\mu }-1\right)$$

As shown in Section A.2, the approximation preserves the mean and $K_{2}$ moment, and the two tail indices of the exact distribution. Figure 1 shows that the above approximation is almost perfect for the PBF distribution. Section A.2 shows cases where the approximation is exact. Figure 2 shows a similar behaviour of the approximation for the Dagum distribution.

Therefore, one can replace the Lorenz curve altogether with three parameters, $K_{2}/\mu ,\xi ,\zeta$. Actually, these three parameters provide much richer information than the Lorenz curve per se. This is illustrated in Figure 3, which compares two distributions with very different behaviour, a PBF and a Dagum, which have the same $K_{2}/\mu$ but different tail indices. It is seen that the Lorenz curves do not give any indication of the different behaviours of the two distributions.

For this reason, we contend that the Lorenz curve, despite its popularity, is not a useful tool to understand the income distribution. A better tool, visualizing the distribution behaviour, is the double logarithmic plot of the odds function, also seen in Figure 3, along with plots of the density function, in linear or logarithmic axes. All three additional plots do not hide the information of the differences, with the logarithmic plots also visualizing the tail indices.

Additional evidence that the Lorenz curve is not a truthful stochastic tool is provided by Figure 4, which is based on actual income distribution data for Bulgaria in 1971. (Nb., we investigate Bulgaria in more detail in Section 4.3.4). The Lorenz curve is smooth and provides no information on the peculiarity of the income distribution in this case. Specifically, the density function plot shows that there is a huge peak at an income slightly lower than the mean, suggesting that most of the population had income close to this value. The Lorenz curve totally hides this fact. Even the K-centre and K-spread plots (lower left panel of Figure 4), while clearly showing the huge departure from the ME exponential distribution, do not provide insight into the extent of the departure from the PBF distribution. On the contrary, the latter is visible in the probability density plot, as well as in the odds function plot. In the latter, it appears as a big plateau at an income slightly lower than the mean, and at odds values around 1.

For these reasons, while for completeness we occasionally show some Lorenz curves in our applications, we do not recommend their use and we strongly propose the odds function double logarithmic plot as a replacement.

For completeness we note that the fitting of the PBF and Dagum distributions, shown in both right-hand panels of Figure 4 and substantially departing from the empirical distribution, was carried out by a least squares method on the odds function. Specifically, the sum of squared differences between the logarithms of the empirical and theoretical odds functions was minimized. The empirical values are readily available given that the data are provided in percentiles of the distribution function. The theoretical values are calculated by the formulae given in Table 1 along with Equation (2). The minimization was performed by a standard nonlinear solver that determined the values of distributional parameters.

The theory of statistics provides several tools to assess the appropriateness of a fitted distribution (e.g., chi-squared test, Kolmogorov–Smirnov test, probability plot correlation coefficient test). Such tests provide quantified validation for the selection of a probabilistic model. However, the scope of our study is not to provide insights into model selection and validation (this topic is covered in the recent (2025) study by Koutsoyiannis [16]); rather, it is to propose and assess indicators of (in)equality and (in)stability based on data of income distribution. For the purposes of our study, graphical tools such as those shown in Figure 4 and subsequent illustrations are more fit to purpose. A big advantage of the plots in both right-hand panels of Figure 4 is that they not only show the inappropriateness of the three theoretical models depicted but they also provide insights on the reasons why the departures of theoretical models appear. In this case, one may diagnose that there was forced equality for the middle class, reflected as a peak in the density function and a plateau in the odds function.

### 3.3. Maximum Entropy Distributions

#### 3.3.1. Unconstrained Bounded Variables

Using calculus of variations, we can determine which is the probability density f(x) that maximizes the entropy, defined in Equation (7), under given constraints. If there is no constraint about the system, apart from the range where the variable lies, specified by the inequality constraint:(32)0≤x≤Ω
then, maximization of entropy results in uniformity, i.e., f(x)=1/Ω, while the maximum entropy, the standardized maximum entropy, and the K-spread coefficient are:(33)$$\Phi \left[\underline{x}\right]=\ln\frac{\Omega }{\lambda },\;\Phi_{\mu }\left[\underline{x}\right]=\ln2,\;\frac{K_{2}}{\mu }=\frac{1}{3}$$

#### 3.3.2. Constrained Mean

However, a system becomes more interesting when, in addition to inequality constraints like (32), or even in their absence, there appear equality constraints, corresponding to the information that is known about a system represented by the variable $\underline{x}$. In studying the material wealth (or income) in a certain society, we assume two characteristic quantities: the mean *μ*, which is related to the total energy available to the society [7], and an upper limit of wealth (or income) Ω, which is mainly determined by the available technology (knowhow) and thus we call it the technological upper limit. One may assert that real income distributions are unbounded from above, but a finite upper limit Ω helps to better understand the framework and may also be useful to model historical situations of the past, in which the technology was elementary. Furthermore the upper limit can easily be removed (and will actually be removed in the next sections) by letting Ω→∞.

The constraints for entropy maximization are thus:(34)$$\int_{-\infty }^{\infty }xf\left(x\right)dx=\mu ,\;0\leq x\leq \Omega$$Assuming a Lebesgue background measure density with $\beta \left(x\right)=1/\lambda$, with λ being a monetary unit (e.g., *λ* = USD 1), the entropy maximizing probability density is [7]:(35)$$f\left(x\right)=\;\frac{1}{\lambda }\frac{e^{-x/\lambda }}{1-e^{-\Omega /\lambda }}$$
which is a (doubly) bounded exponential distribution. The particular characteristics of the distribution are given in Table 1. Illustrations of the density function for two values of the upper bound Ω are seen in Figure 5 (left). In addition, Figure 5 (right) shows the variation of the mean, K-spread coefficient, and standardized entropy, with the upper limit Ω, standardized by the scale parameter λ. All three quantities increase with the increase of Ω. As Ω/λ→∞, the K-spread coefficient tends to the value $K_{2}/\mu =1/2$ and the standardized entropy tends to $\Phi_{\mu }=1$.

As the tendency of entropy is to grow, one may understand that human societies would push the technological limit to high values and this has actually happened historically [7]. In other words—and despite the bad name of entropy because of misunderstanding its meaning—the tendency of entropy to become maximal is the agent of change and technological progress. As seen in Figure 5 (right), once the technological limit became high enough, say Ω/λ≈10, it could be neglected as if Ω/λ=∞. In this case we obtain the standard (unbounded) exponential distribution, also shown in Table 1. The characteristics of the latter distribution, namely K-spread coefficient $K_{2}/\mu =1/2$ and standardized entropy $\Phi_{\mu }=1$, define a characteristic point or a *pole* on a 2D plane ($K_{2}/\mu ,\Phi_{\mu }$), which (provided that our variable is nonnegative) cannot be surpassed in the sense that $\Phi_{\mu }$ cannot take any value higher than 1, and the highest value of 1 can be achieved only if $K_{2}/\mu =1/2$, otherwise it would necessarily be smaller. Hence the distance of a specific country’s state from this pole, i.e.,(36)$$d_{p}:=\sqrt{{\left(\frac{K_{2}}{\mu }-\frac{1}{2}\right)}^{2}+{\left(\Phi_{\mu }-1\right)}^{2}}$$
is a useful index for the characterization of an economy’s state, additional to those already discussed.

It can be shown (and confirmed in Figure 5, right) that, as Ω/λ→0, the mean μ tends to Ω/2, the K-spread coefficient tends to the value $K_{2}/\mu =1/3$, and the standardized entropy tends to $\Phi_{\mu }=\ln2$. This signifies a uniform distribution (see Equation (33)). Furthermore, even though not shown in Figure 5, Equation (35) allows for values λ<0 and in this case we have the (bounded) anti-exponential distribution. As Ω/λ→−∞, the mean μ tends to Ω, the K-spread coefficient tends to the value $K_{2}/\mu =0$, and the standardized entropy tends to $\Phi_{\mu }=-\infty$. This signifies a distribution with all probability mass concentrated at x=Ω=μ=0 (an impulse). The impulse represents certainty, with full equality of the population in economic terms. Stochastically, the reason for the emergence of these types of distribution, which must have been materialized in the far distant past at the cradle of human societies, is the very low technological limit, which did not allow any options for diversity in income. Interestingly, Marx and Engels [17] and their followers interpreted this situation of misery as representing an ancient classless society, and envisaged recreating it in the future.

#### 3.3.3. Constrained Mean and K-Spread

For the next step, we pose an additional constraint, namely of a fixed K-spread coefficient, or equivalently a fixed $K_{2}$ moment, also removing the upper limit. In this case, the determination of the resulting entropy maximizing distribution is cumbersome and is given in Section A.3. The result, if the domain of the variable is the entire line of reals, is the logistic distribution:(37)$$F\left(x\right)=1-\frac{1}{1+e^{-\varsigma +x/\lambda }}$$
where ς is a parameter. If $\underline{x}\geq 0$, as in the case of income, the resulting distribution is the generalized half-logistic (GHL), whose expression is:(38)$$F\left(x\right)=1-\frac{1}{1-e^{-\varsigma }+e^{-\varsigma +x/\lambda }}$$

It can be easily verified that, if $\varsigma =0\;(and\;so\;e^{-\varsigma }=1)$, in the case of Equation (37), we get the standard logistic distribution, while, in that of the Equation (38), the distribution becomes identical to the exponential one. Thus the GHL distribution contains the exponential distribution as a special case.

The details of both distributions are contained in Table 1. In the case of GHL, the equation giving the K-spread coefficient, albeit simple, cannot be solved explicitly for the parameter ς; yet, if $K_{2}/\mu$ is known, it is easy to find ς numerically and then determine the standardized entropy from ς. A good analytical approximation of ς is given by:(39)$$\varsigma \approx \left\{\begin{array}{ll} \frac{0.975}{K_{2}/\mu }-5.85K_{2}/\mu +1 & 0<K_{2}/\mu <1/2\\ \frac{0.975}{K_{2}/\mu -1}-5.85\left(K_{2}/\mu -1\right)-1 & 1/2<K_{2}/\mu <1 \end{array}\right.$$
and of standardized entropy by:(40)$$\Phi_{\mu }\approx 1-6{\left(\frac{K_{2}}{\mu }-\frac{1}{2}\right)}^{2}-\frac{1}{3}{\left(-\ln\left(1-4{\left(\frac{K_{2}}{\mu }-\frac{1}{2}\right)}^{2}\right)\right)}^{\frac{5}{3}}$$

The thus established relationship between $K_{2}/\mu \;and\;\Phi_{\mu }$ is illustrated in Figure 6 (curve named “exact ME GHL distribution”), which also shows the satisfactory behaviour of the approximation in Equation (40) (curve named “explicit ME GHL approximation”).

This relationship represents the trade-off between entropy and the K-spread coefficient (Gini index). As $K_{2}/\mu$ departs from the pole (point (1/2, 1)), the entropy decreases in a symmetrical manner around the line $K_{2}/\mu =1/2$. Points below the curve of Figure 6 are mathematically (and practically) feasible, while those above the curve are infeasible. Values $K_{2}/\mu <1/2$ indicate low stratification of society in terms of income, and values $K_{2}/\mu >1/2$ indicate high stratification. As per entropy, we have (arbitrarily) partitioned the area below the curve of Figure 6 into three parts, based on the distance from the pole $d_{p}$. Values of $d_{p}$ between 2/3 and 1 indicate high stability of economy, those between 1/3 and 2/3 indicate low stability, and those below 1/3 indicate high instability. The latter area also includes negative entropy values, which in reality are feasible but not quite common.

It is useful to approximate the curve of Figure 6 with more general distributions, such as PBF and Dagum, as well as special cases thereof, such as Pareto and Weibull. As seen in Figure 6, which also compares the exact and approximate curves, and in Figure 7 (left), which shows the approximation errors, all four distributions provide good approximations. Figure 7 (right) shows that, in the approximating distributions, the tail indices are no longer ξ=0, ζ=1 but vary depending on the K-variation coefficient.

Most promising are the approximations by the two-parameter distributions Weibull and Pareto. The Weibull distribution provides good approximation for the low stratification part of the curve ($0<K_{2}/\mu <1/2$). The upper tail index is ξ=0, while the required value of the lower tail index ζ and the achieved standardized entropy $\Phi_{\mu }$ are:(41)$$\zeta =-\frac{\ln2}{\ln\left(1-K_{2}/\mu \right)},\;\Phi_{\mu }=1+\left(1+\frac{\ln\left(1-K_{2}/\mu \right)}{\ln2}\right)\gamma -ln\Gamma \left(-\frac{\ln\left(1-K_{2}/\mu \right)}{\ln2}\right)$$

The Pareto distribution provides good approximation for the high stratification part of the curve ($1/2<K_{2}/\mu <1$). The lower tail index is ζ=1, while the required value of the upper tail index ξ and the achieved standardized entropy $\Phi_{\mu }$ are:(42)$$\xi =2-\frac{1}{K_{2}/\mu },\;\Phi_{\mu }=3-\frac{1}{K_{2}/\mu }+\ln\left(\frac{1}{K_{2}/\mu }-1\right)$$

#### 3.3.4. Notes on the Tail Indices

As will be seen in the applications (Section 4), while the curve shown in Figure 6 effectively captures the feasible space of the covariation of entropy and K-spread, the underlying GHL distribution proves not appropriate for modelling the income distribution. The main obstacle is its tail indices, which are ξ=0, ζ=1, same as in the exponential distribution. In reality, however, any deviation from the exponential distribution is due to different tail indices rather than due to a different K-spread with the same tail indices.

In all applications, the lower tail index ζ is always >1 and the upper tail index ξ is always >0. Hence neither the exponential nor the GHL distribution can accurately model reality. Moreover, these two tail indices are too important to ignore in economic studies. In fact, there are good reasons that they differ from their ME values [7]. A ζ>1 reflects the role of a state in an organized society to redistribute income and wealth through their transferal from richer individuals to poorer by means of several mechanisms, such as taxation, public services, land reform, monetary policies, and others. On the other hand, ξ>0 reflects the politico-economic power of the richest, who pursue a greater share of the community’s wealth, thus tending to modify mostly the income distribution tail, converting it from exponential to power-law. At the same time, this advances both the technological limit and the average wealth—a positive side of the elites’ actions for the entire society.

One would think about imposing additional constraints that would modify the distribution from exponential to power law with ξ>0,ζ>1. This is not so easy, but adopting a non-Lebesgue background measure density $\beta \left(x\right)$ makes it easier without imposing additional constraints [15]. With a proper background measure, the resulting ME distribution becomes PBF or Dagum. If we adhere to the Lebesgue measure, then neither of these two distributions is an entropy maximizing distribution, as it can be shown that their density functions are not even local (let alone global) maximizers of entropy under specified mean and K-spread coefficients.

Nevertheless, both these distributions can provide high standardized entropies, depending on the specific values of the indices ξ,ζ, and could approximate with high accuracy even the GHL distribution as discussed in Section 3.3.3. Yet, the total effect of specifying values of ξ>0 or ζ>1 or both is the shift of the entropy curve lower than that of GHL. Some examples of this effect are shown in Figure 8.

The PBF and Dagum distributions are both related and complementary to each other. Obviously, one of the two would provide a better fit to the empirical distribution than the other and, depending on the specific values of the indices ξ,ζ, will give higher entropy than the other. The areas of the combinations of parameter values that lead each one to yield higher entropy than the other are depicted in Figure 9, constructed after a systematic numerical investigation. Details on the information provided by the figure are shown in its caption.

In many applications, the PBF and Dagum distributions are thought to lead to overfitting, especially when dealing with limited observed samples, and the estimates of their tail indices are regarded as too uncertain. However, this is not the case with the data used here, as they are summarized data from huge datasets, provided in terms of percentiles. As already discussed, the tail indices are very important and cannot be neglected, while, additionally to those, the two distributions have only one scale parameter, which is again necessary to consider. Hence these distributions are not over-parametrized, but rather have a minimal parameter set. Moreover, the estimates of the tail indices can be carried out even without fitting a probabilistic model by means of the empirical odds function using Equation (25) and determining the log-log slopes with the first or last handful of data points. Actually, the values provided in the application below were estimated in this way and later confirmed by fitting the PBF and Dagum models.

## 4. Application

### 4.1. General Setting

Here we apply the framework theoretically developed in Section 3 to the income data of several countries and different periods. This application has two parts. In the first we examine the characteristics of the countries with large populations—a total of 17 countries with populations over 50 million and with data availability for the year 2022, which is the reference year. Our aim here was to use the most recent year, which in the database is 2023, but there were too many missing data and we preferred to use 2022. In addition, we compiled merged datasets for two multi-country entities, namely the European Union (27 countries) and the World (69 countries, for which data from 2022 are available, with a total population of 5.4 billion). The compilation process is described in Section B.1. The total of 19 entities examined include major geopolitical powers, such as the USA, China, India, the Russian Federation, and the European Union, belonging either to the G7 or the BRICS group. Other countries of these groups (e.g., France, Germany, the United Kingdom for G7) are also regarded by many as geopolitical powers, but the results of our analyses (and many more indications) will not confirm such a characterization. In any case, such characterization is subjective and time changing, and does not affect the objective results of our analyses.

In this second part, we examine the history of the evolution of two major economic indices, the K-spread (Gini index) and standardized entropy, as well as the distance from the pole of maximum entropy for four countries, in correlation with the political history of each of them. These countries were chosen for the peculiarities in their politico-economic evolution, provided that their data also exhibit sufficient historical depth. Specifically, Argentina (1953–2023) was selected as a case study of a country that experienced successive military coups and political turmoil during the period 1953–1980. Bulgaria was chosen as an example of a Soviet satellite state attempting to implement the communist system and an egalitarian income distribution. Brazil (also included in the first group) was included as a country that, despite its severe inequalities, has historically recorded Gini values near 1/2. Finally, South Africa (also included in the first group) was selected due to its pronounced economic disparities and its persistent and extremely rigid social stratification.

A brief political overview of each country under examination is provided, beginning with a brief history and the political landscape to the time series examined. This allows us to form a concise understanding of broader societal perceptions regarding social stratification in different regions. Even if the available data for our analysis refer mainly to the second half of the 20th century and the early years of the 21st century, with these tools, we are trying to evaluate the social dynamics and the historical evolution of each country.

In all cases, from the available data, all information referred to in Section 3 was extracted but presented only partly to avoid a very long text. The processing of the data is described in Section B.2 for the assignment of empirical values of the distribution function and density function for observed values of the income (given in percentiles), and in Section B.3 for the calculation of empirical values of entropy. The PBF and Dagum distributions were fitted in all cases, and the numerical values of indices were calculated both directly from the data (as already described in Section 3.3.4) and indirectly from the fitted distribution. The direct and indirect values were very similar and hence only the former are reported here. We note though that, in extreme cases of large deviations of data from the two distributions, like in Bulgaria, 1971, shown in Figure 4, the direct empirical values differ from the indirect ones.

### 4.2. The Status of the Major Countries in 2022

In 2022, countries with populations over 50 million, and in particular major geopolitical powers, exhibited distinct perceptions of inequality shaped by historical legacies, economic structures, and policy approaches, influencing their social and political landscapes [18,19]. The United States framed inequality as a consequence of market-driven innovation, with policymakers often downplaying wealth concentration among the top 1% while public discourse highlighted racial and economic divides, amplifying polarization [20,21]. On the other hand, China’s leadership viewed inequality as a manageable byproduct of rapid growth since the 1978 reforms, prioritizing urban development and poverty reduction while tolerating wealth concentration among elites, with the “common prosperity” initiative signalling a shift toward addressing urban–rural disparities through targeted redistribution [22,23].

India perceived inequality as a structural challenge rooted in colonial land systems and informal economies, with elites accepting stark wealth gaps as a trade-off for growth, though public frustration over stagnant wages and urban–rural divides fuelled demands for reform without cohesive policy action [24,25]. The European Union saw inequality as a threat to social cohesion, emphasizing robust welfare systems and progressive taxation to ensure equitable income distribution, though regional variations and crises like inflation sparked debates over deeper fiscal unity [26]. Russia perceived inequality as secondary to state stability, with economic measures boosting lower incomes but elite wealth concentration and regional disparities accepted as entrenched features of its resource-driven system [27].

Detailed graphs of the economic status (similar to those for Bulgaria in Figure 4) are given in Figure 10 for the USA, in Figure 11 for China, and in Figure 12 for the World. The graphical depictions for the USA and China show a close similarity between the two cases, with the most visible difference being the smaller upper tail index of China, visualized by the slope of the rightmost part of the odds function curve.

On the other hand, the graphs for the composite sample of the World show distinct differences from those of USA and China, with much greater inequalities, mostly reflected in the K-spread profile, which is higher than that of the exponential distribution. In all cases, though, the PBF and the Dagum models provide good fits to the empirical distributions.

The K-spread vs. standardized entropy curve for all 19 entities examined are shown in comparison to each other and to the GHL curve in the upper left panel of Figure 13, while in the upper right graph the distances from the pole of maximum entropy are compared. For completeness, the same graph in its lower panels provides information on the gross domestic product (GDP) per capita and gross domestic product based on purchasing power parity (GDP-PPP) per capita. These are important indices of prosperity, but they were not investigated in detail here, as the focus is on (in)equality and (in)stability of economy.

The composite case of the World appears to have higher entropy than individual countries, which is expected because it incorporates many very different economic models, leading to high composite uncertainty. Observing the position of the geopolitical players in 2022 from the perspective outlined in the methodology above, we see that the K-spread coefficient (Gini index) does not reflect China’s political intentions regarding common prosperity, since it appears rather high. In contrast, evaluation through entropy captures the social dynamics more accurately, as China appears to be the most stable of all individual entities (just below the World), exhibiting the smallest distance from the pole of maximum entropy. India, although positioned according to the Gini index as having the potential to achieve maximum entropy, does not succeed in remaining close to the pole. The United States is at a greater distance than China and India, yet still within a stable framework; the European Union and Russia are located close to the boundary of high stability, while the United Kingdom, France, and Germany lie at the area of low stability. Except for these three, all other countries and composite entities examined lie in the high stability area.

In addition, Table 2 summarizes the main numerical indices resulting from this analysis. The lowest or highest values of the indices that favour equality or stability are highlighted in bold and it can be seen that China and Russia are the most notable in good performance in this respect, with Italy and Germany (in terms of equality) following. At the other end, of not good performance are Germany (in terms of stability), India, Turkey, and South Africa.

### 4.3. A Brief Political History and the Evolution of Economic Indices in Specific Countries

#### 4.3.1. Argentina

Argentina’s political history began with independence from Spain in 1816, leading to a period of civil wars between federalists and unitarians, eventually stabilizing under a federal constitution in 1853. The late 19th century saw economic prosperity driven by agricultural exports and European immigration but also rising social tensions and oligarchic rule, culminating in the 1916 introduction of universal male suffrage and the Radical Civic Union’s ascent. The 1930 military coup marked the start of instability, followed by the 1943 coup that propelled Perón to power in 1946, establishing Peronism as a transformative force blending populism and authoritarianism [30].

In the 1950s, Argentina’s political and social perceptions were profoundly shaped by Peronism, a dominant political culture, emphasizing social justice, economic independence, and political sovereignty as a “third way” between communism and capitalism. Argentines under Peronism sought a balanced system that incorporated elements of state intervention in the economy without full communist collectivization, fostering a populist identity that prioritized national sovereignty over ideological extremes [31,32].

From the 1950s to the 1980s, Argentina experienced recurrent military interventions, including the 1955 coup that ousted Perón, the 1962 and 1966 coups against civilian governments, and the 1976 coup that initiated the brutal Dirty War under a military junta [33]. This era saw alternating periods of restricted democracy and outright dictatorship, and social unrest (Figure 14). The frequent upheavals stemmed from deep-seated political polarization, economic volatility including hyperinflation and debt crises, and a tradition of military involvement in politics [34].

Since the return to democracy in 1983, Argentina’s political history has been characterized by efforts to consolidate democratic institutions amid economic challenges [35]. In the 1990s, neoliberal reforms were introduced, followed by the 2001–2002 economic collapse and a series of short-lived presidencies. The period 2003–2015 saw progressive policies and debt restructuring, followed by shifts toward market-oriented reforms [36] and libertarian austerity measures, reflecting ongoing struggles for stability in a polarized landscape [37].

Observing Argentina’s history from the perspective outlined in the methodology above, we can see that the Gini index does not provide us with information regarding the stability and condition of social inequalities. Before 1980, this index ranged between 0.32 and 0.41, with an average value of 0.35; during 1980–1999, it ranged between 0.38 and 0.48, with an average of 0.44; and in the period since 2000, it ranged between 0.38 and 0.50, with an average of 0.42. Thus, if we were to evaluate social status solely through this index, we would conclude that the period before 1980 was of greatest social harmony—something that clearly did not occur.

**Figure 14 entropy-28-00035-f014:**
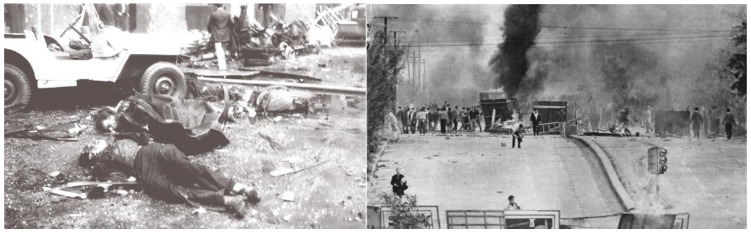
Characteristic violent events in Argentina in the 20th century: (**left**) civilian casualties after the air attack and massacre on Plaza de Mayo, June 1955 [38]; (**right**) Cordobazo general strike in protest against the political and economic decisions of the military dictatorship, Bulevar San Juan, Córdoba Capital, May 1969 [39].

On the contrary, if we examine entropic indices, we observe that, before 1980, the entropy ranged between −0.12 and 0.59, with an average of 0.25; during 1980–1999 between 0.69 and 0.78, with an average of 0.74; and since 2000 between 0.71 and 0.82, with an average of 0.77. Clearly, in the period before 1980, the annual entopic indices lie in the areas of low stability to high instability, thereby explaining the social instability, the unrest, and the coups of that period, since the social structure was fragile—something not reflected in the Gini index (Figure 15 left).

By combining the two indicators, using the distance from the pole of maximum entropy as the indicator of stability, we find that the distances of the distributions before 1980 range between 0.44 and 1.12, with an average of 0.76; during 1980–1999 between 0.23 and 0.32, with an average of 0.27; and since 2000 between 0.19 and 0.30, with an average of 0.24. Taking into account that a smaller distance from the pole of maximum entropy indicates greater stability in the distribution, the large distances once again of the period before 1980 explain the upheavals and coups (Figure 15 right).

#### 4.3.2. Brazil

Brazil’s political history began with independence in 1822 as a monarchy transitioning to a republic in 1889 amid the abolition of slavery and coffee boom-driven growth [40]. The Old Republic (1889–1930) was dominated by oligarchs, followed by the authoritarian Estado Novo (1937–1945), which introduced labour rights but centralized power [41]. Post-1945 democracy was interrupted by the 1964 military coup, establishing a dictatorship until 1985 that accelerated industrialization but deepened inequalities through repression and economic policies favouring elites.

Brazil’s political perceptions have been shaped by a legacy of colonialism, slavery, and elite dominance, leading to extreme inequalities rooted in the latifundia system where large landowners controlled vast estates, exploiting labour without significant redistribution. Favelas (Figure 16) emerged in the late 19th century as informal settlements for freed slaves and rural migrants, exacerbated by urbanization without social reforms [42]. Such inequalities persisted without major revolutions due to a tradition of clientelism, military repression, and co-optation of dissent through gradual reforms, preventing widespread uprisings despite stark disparities [43].

From 1980 to the present, Brazil transitioned from military rule [45] to democracy, with the 1988 Constitution emphasizing social rights [46]. The 1990s focused on economic stabilization via the Real Plan, reducing inflation. The 21st century was marked by attempts for poverty reduction coexisting with corruption scandals, political polarization, and ongoing challenges like inequality and environmental issues [47].

Observing history after 1980 (when data are available), from the perspective described in the methodology above, we see that the Gini index ranges roughly between 0.48 and 0.58, with a mean of around 0.53, higher than in the other countries examined above yet not reflecting extreme inequalities. However, the upper tail index ξ (not shown in the graphs) has a very high value, more than 0.5 and reaching 0.6, and this better reflects the fact that the social stratification system is far from ideal, as also depicted in the favelas and elsewhere. It appears that the inequalities stem from the political–cultural legacy of Brazil’s earlier colonial era, and that modern policies have not been able to fully eradicate those practices. If we examine social stability from the viewpoint of entropy, we find that it ranges between about 0.77 and 0.87, except for year 2005, when it was 0.63 (Figure 17). Generally, this level does not indicate political instability, and indeed no major instability has been observed in the period under review. Notably, in the year 2005, a major political scandal (the Mensalão scandal) broke out, in which the ruling Workers’ Party was accused of monthly payments to deputies to vote as the government wished [48].

The scandal had institutional consequences: public pressure, resignations of senior government officials, and suspicions of broader corruption. Although Brazil’s economy at that time did not collapse, the entropy derived from the income distribution registered an impressive drop, signifying that a destabilization event did occur—one that was later rectified.

#### 4.3.3. South Africa

South Africa’s political history involved Dutch settlement in 1652, British conquest in the early 1800s, and the 1910 Union formation excluding Black participation. The National Party’s 1948 victory institutionalized apartheid, enforcing racial separation and suppressing resistance through events like the 1960 Sharpeville Massacre and 1976 Soweto Uprising. International sanctions and internal protests in the 1980s eroded the regime, leading to 1990 reforms and initiating negotiations [49,50].

South Africa’s political perceptions were shaped by centuries of colonialism and racial domination, with extreme inequalities stemming from the exploitation of Black labour under Dutch and British rule, formalized in apartheid from 1948. This tradition of segregation, including land dispossession and pass laws, created vast disparities without immediate overthrow due to military suppression and divide-and-rule tactics [51]. The anti-apartheid struggle culminating in the 1990–1994 transition, with Nelson Mandela’s release and the 1994 democratic elections marking the end of white minority rule [52].

From 1990 to the present, South Africa transitioned to democracy with Mandela’s 1994 presidency, focusing on reconciliation via the Truth and Reconciliation Commission and affirmative action. The 21st century developments are not free of corruption scandals [53], while inequalities persist, fuelling protests [54,55].

Observing history from the perspective of our methodology, and in particular the evolution of the Gini index, we see that, although apartheid was overturned in the period under examination (after 1993), inequalities in South Africa continued to be extremely large—with the index fluctuating between 0.67 and 0.74 and an average of 0.70 (Figure 18). It appears that, in South Africa, the intense inequalities stem from the political–cultural legacy of earlier eras, while modern policies have not managed to smooth them out. If we examine social stability from the lens of entropy, we find that it ranged between 0.51 and 0.72, with a mean around 0.62, indicating the country is in a state of low stability, as also suggested by high corruption indices [56] and elevated crime rates [57,58,59].

#### 4.3.4. Bulgaria

Bulgaria’s modern political history started with its independence in 1878 after centuries under Ottoman control, followed by a monarchy and participation in the Balkan Wars. Its alignment with the Axis in WWII led to Soviet occupation in 1944, installing a communist regime under Georgi Dimitrov by 1946, with purges and nationalization. Todor Zhivkov’s long rule from 1954 emphasized Soviet loyalty, culminating in 1989 protests and the regime’s fall amid perestroika [60,61].

Communism arose through Soviet imposition rather than a strong indigenous tradition of equality or social stratification redistribution, though some agrarian reforms built on pre-existing peasant movements. There was no deep-rooted culture of egalitarian distribution, as the system was enforced top-down amid purges and collectivization. Figure 19 shows the architectural expression of the communist era, when all houses were quite similar, in huge blocks.

From 1960 to 1989, Bulgaria pursued industrialization and cultural assimilation policies. The 1989 ouster led to multiparty elections in 1990, with socialist governments initially dominating the transition to market economy, followed by EU accession in 2007 [62].

Observing history from the perspective of our methodology, we see a very low Gini index during the period 1963–1989, ranging between 0.18 and 0.26, with an average of 0.23 (Figure 20). This means that a high level of equality was achieved, according to the goals of the communist regime. In the period 1992–2023, when Bulgaria entered the free market, inequalities amplified and the Gini index increased to 0.31–0.42, with an average of 0.36.

From the perspective of entropy, we observe that during 1963–1989 it ranged between −0.67 and 0.39, with an average of 0.12, whereas during 1992–2023 it ranged between 0.60 and 0.80, with an average of 0.72. It is noteworthy that negative entropy values appear in several years of the communist period, with the lowest value in 1971, a turbulent period for Bulgaria (Figure 20, left) [63]. Indicatively, on 16 May 1971, the referendum on the Zhivkov Constitution was held, with voter turnout at 99.7% and approval also at 99.7% [64]. These extraordinarily high rates suggest that political practices were democratic only in appearance and the system had limited real alternatives. This manifests as a strikingly low entropy of the social structure of that time.

**Figure 19 entropy-28-00035-f019:**
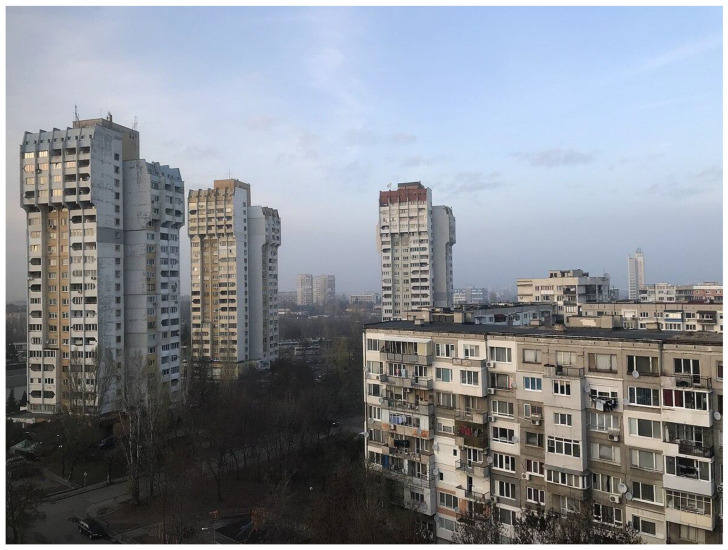
Apartment blocks in Sveta Troitsa, Sofia, Bulgaria, at the northwest end of the district, next to a train station [65].

**Figure 20 entropy-28-00035-f020:**
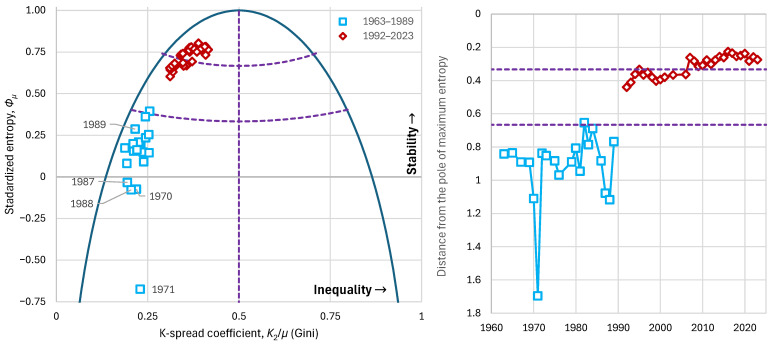
Characteristic graphs for the evolution of major economic indices in Bulgaria: (**left**) standardized entropy vs. K-spread coefficient (Gini index), plotted alongside the maximum entropy vs. K-spread curve; (**right**) distance from the pole of maximum entropy. The cyan rectangles represent the era of Soviet influence and the dark red diamonds the era of free market. Purple dashed lines show the boundaries between the partitioned areas.

It is interesting that almost the entire communist period falls within the area we have characterized as one of high instability, with notable episodes such as in 1971, already mentioned above, as well as the years just before the collapse of the Soviet Union (1987–1988), during which the system also exhibits negative entropy and large distance from the pole of maximum entropy (Figure 20, right)—something that explains the eventual overturning of this political system.

## 5. Discussion and Conclusions

Entropy carries a bad reputation in both scientific and public discourse [7], but this can be attributed to the fact that its meaning is greatly misunderstood because it is a stochastic concept, while the education system is based on the deterministic paradigm. Far from signifying decay, decadence, or disorder as usually thought, entropy is a formal quantification of uncertainty, the dominant feature in complex real-world systems. The tendency of entropy to increase and the related principle of maximum entropy formally describe the natural tendency of complex systems to move from less probable to more probable states. High entropy corresponds to a greater multiplicity of states, hence expanded freedom of choice, more opportunities, and structural resilience.

Being a non-conservation law, entropy maximization is also a driver of change. This is also the case in economics and we have shown that, starting from a bounded distribution that has low entropy, the inevitable tendency of entropy to grow would push the technological limits to high values—a pattern historically confirmed. Technological progress as well as growth of wealth are not merely compatible with entropy increase, they are its direct expression.

The typical tools used in economic analyses, namely the Lorenz curve and the Gini index, totally miss accounting for entropy. Here we showed that Lorenz profiles are a poor representation of the economic states and hence we recommend replacing them with simpler graphs such as the odds and probability density functions. The Gini index, which we showed is identical to the (second-order) K-spread coefficient, $K_{2}/\mu$, is a good indicator of (in)equality but neglects dynamics in distribution tails. Therefore, we propose complementing it with upper and lower tail indices, ξ, ζ and also accompanying it with a standardized form of entropy, $\Phi_{\mu }$.

We also demonstrated here that, under constraints of specified mean, μ, and K-spread, $K_{2}/\mu$, the maximum entropy distribution is the GHL distribution, a limiting case of which is the exponential distribution. The latter materializes the peak entropy pole, as ($K_{2}/\mu =1/2,\Phi_{\mu }=1)$. The limiting curve of $\Phi_{\mu }$ vs. $K_{2}/\mu$, or else the maximum entropy vs. K-spread curve, turns out to be a parabola-like shape symmetrically arranged below this pole. The distance from this pole is another indicator of resilience or stability of an economy, with a small distance denoting small instability.

The real-world applications with data (percentile records) from the World Income Inequality Database, illustrated the theoretical framework and provided support to its hypotheses and results. The country-level analyses (Argentina, Brazil, South Africa, Bulgaria) showed that entropy declines can be linked to political ruptures. In addition, the analyses of the core set of indicators in present-day geopolitical powers (China, India, USA, Russia, the EU) affirm their stability based on the criteria developed, even though some EU countries lie in the low stability area. Interestingly, in all latter cases, the K-spread index is lower than 1/2, positioning these geopolitical powers to the low stratification area of the maximum entropy vs. K-spread graph.

High stratification is rarer, but it was affirmed in the case of South Africa, where in recent years a tendency to increased entropy is noted, albeit without one to decreased stratification. In contrast, very low stratification, quantified by the K-spread coefficient, was the case in former socialist countries, of which Bulgaria was studied in detail. Interestingly, even in this case, higher order spread parameters, such as $D_{10}/D_{2}$ kept high values, despite the low $K_{2}/\mu$. Naturally, the entropy in this period was too low, placing the country in high instability. This radically changed after the fall of the communist regime, with the entropy substantially increasing, thus leading to higher stability.

Apparently, entropy, K-spread, and the other indices studied do not provide a complete picture of prosperity. Absolute indicators such as the GDP per capita and the GDP-PPP per capita should also be considered, but they were not the focus of this study—even though we also provided these indicators for the entities examined. Indices of “real economy” (dealing with goods and services that satisfy human needs and desires, such as agriculture, manufacturing, construction, and services), as contrasted with the “financial economy” (dealing with financial assets like stocks and bonds), are also most important but outside the scope of this study. Societal aspects such as equal opportunities, freedom of choice, and creative expression, and ultimately a meritocratic structure that would not be influenced by hereditary or entrenched class constraints, are also important drivers of economy. Our data do not allow us to make this kind of approach, but it would be interesting to explore it in future research.

The findings of our study, and in particular the proposed indicators (K-spread, standardized entropy, tail indices) and the concave frontier between entropy and K-spread, may have practical implications. Yet we avoid discussing them or providing policy recommendations, actionable insights for policymakers, or usable information for policy decisions, such as tax reforms, social welfare programs, or regulatory interventions. We prefer to adhere to the scientific part of the subject matter, leaving the study of such implications to policy experts.

Although our analysis focused on the country and international level, the proposed methodology could be applied at smaller scales, such as in businesses, organizations, or local communities. In these contexts, calculating similar indicators of relevant variables, e.g., wage distribution within a company, could provide valuable insights into the quality, effectiveness, and stability of the policies implemented.

The country-level analyses revealed that, while the maximum entropy vs. K-spread curve is a tool of high explanatory potential, the underlying GHL distribution is hardly representative of the actual statistical behaviour. Its specified tail indices at ξ=0, ζ=1 do not correspond to real situations where both tail indices turn out to be higher than the GHL values. Therefore, our framework included the flexible PBF and Dagum distributions, which usually had excellent performance in terms of fitting in real-world data. Yet, there is space for future research with constraints different from a specified K-spread coefficient, or with varying background measures, which would result in better agreement between theory and real-world data.

Hopefully, our framework transforms inequality analysis: entropy is not a penalty on growth but its engine. By embracing uncertainty as freedom, we reconcile equity with innovation—a synthesis that Aristotle intuited: virtue lies in the mean but excellence in the extreme.

## Figures and Tables

**Figure 1 entropy-28-00035-f001:**
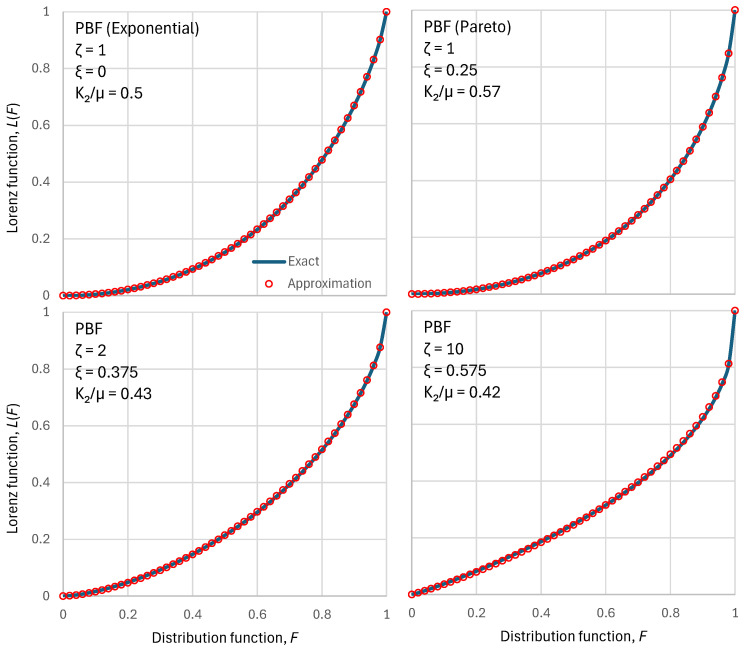
Visual comparison of the exact Lorenz curves for the PBF distribution function with four different parameter sets (shown in each of the four panels) with the approximation of Equation (30). Except for the Pareto case, the ξ parameter was determined to maximize entropy for a chosen ζ parameter.

**Figure 2 entropy-28-00035-f002:**
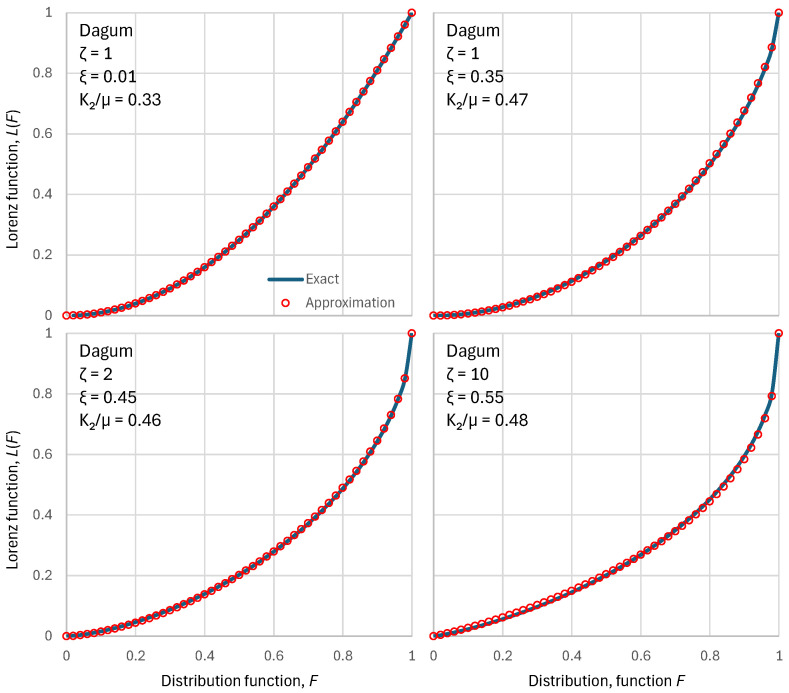
Visual comparison of the exact Lorenz curves for the Dagum distribution function with four different parameter sets (shown in each of the four panels) with the approximation of Equation (30). Except for the upper-left case, the ξ parameter was determined to maximize entropy for a chosen ζ parameter.

**Figure 3 entropy-28-00035-f003:**
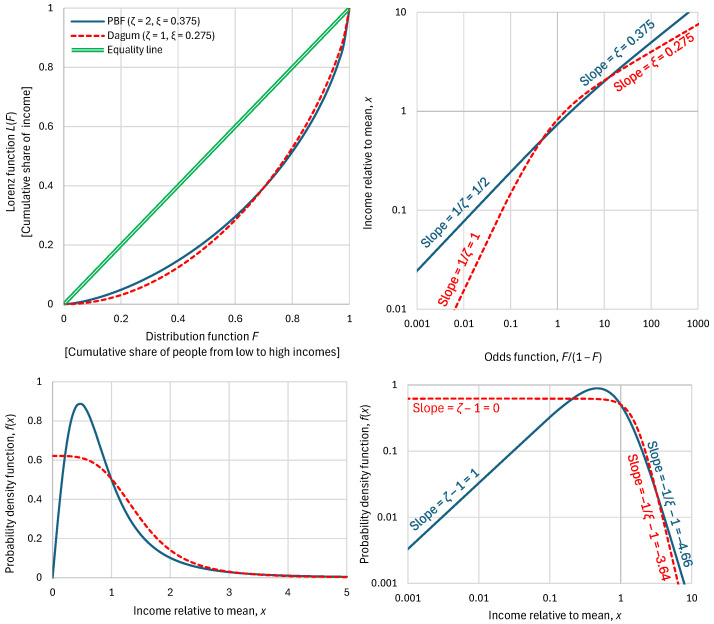
Visual comparison of statistical characteristics of two distribution functions, PBF and Dagum, with parameters as shown in the legend, having the same K-spread coefficient (Gini index), $K_{2}/\mu =0.43$, and different standardized entropies $\Phi_{\mu }=$ 0.88 and 0.97, respectively. (**Upper left**) Lorenz curves, which are very similar for the two distributions; (**upper right**) distribution functions plotted in the form of the variable x vs. the odds function $F(x)/(1-F\left(x\right))$, which shows the substantially different behaviour of the distributions, especially in the tails; (**lower**) probability density function on (**left**) linear and (**right**) logarithmic plots.

**Figure 4 entropy-28-00035-f004:**
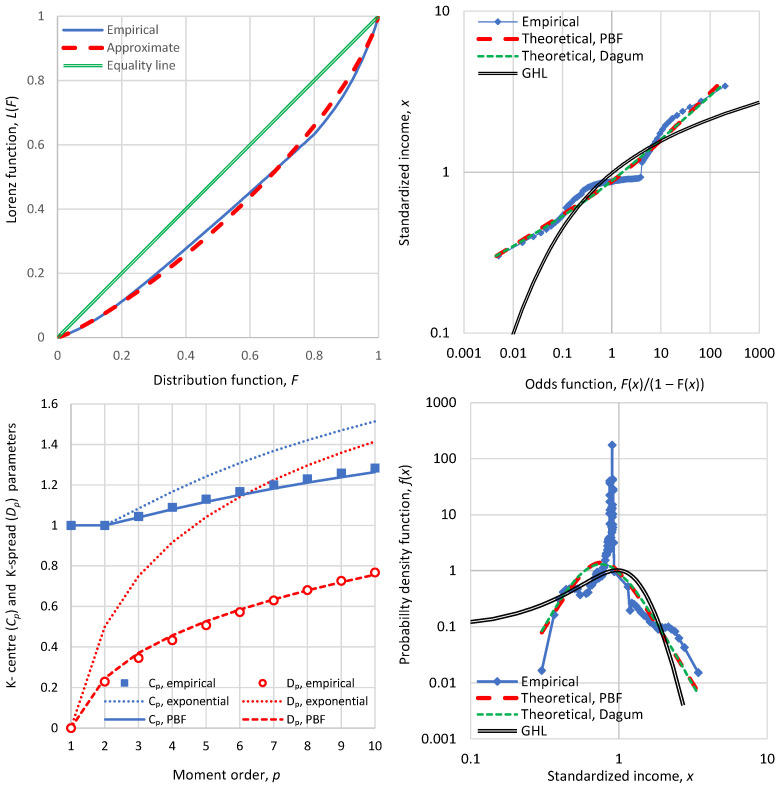
Detailed graphs of economic indicators of Bulgaria in 1971 (**clockwise from upper left**): Lorenz curve; odds function; probability density function; and graph of K-centre and K-spread vs. K-moment order, where for reference the theoretical curves of the ME exponential distribution are also plotted in dotted lines.

**Figure 5 entropy-28-00035-f005:**
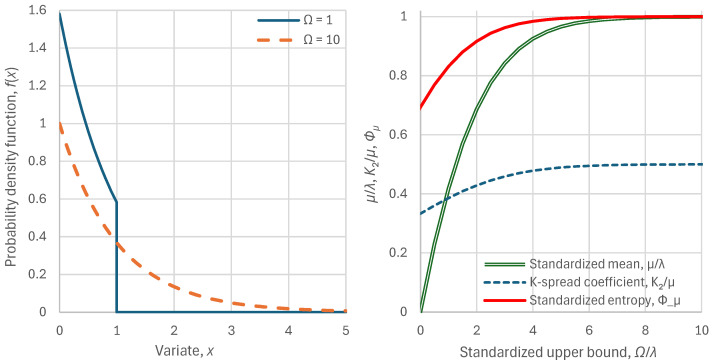
Visualization of the behaviour of the bounded exponential distribution, resulting from maximization of entropy for fixed mean μ and upper bound Ω: (**left**) two instances of the probability density function for the indicated values of the upper bound and for scale parameter λ=1; (**right**) variation of standardized mean, K-spread coefficient (Gini index) and standardized entropy, for varying upper bound Ω/λ.

**Figure 6 entropy-28-00035-f006:**
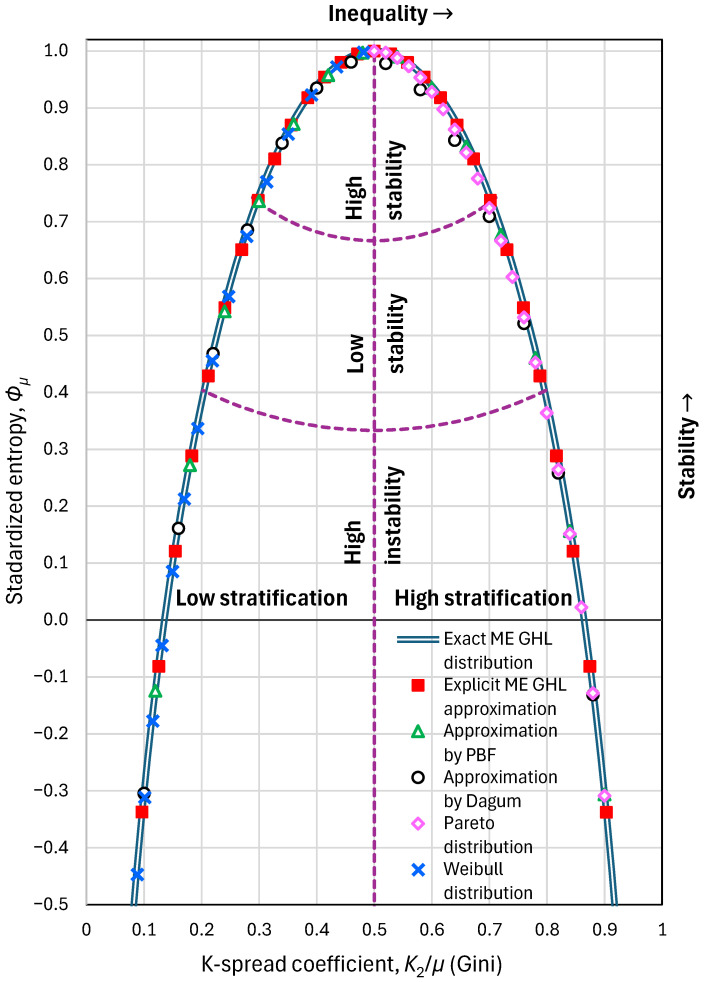
Maximum entropy vs. K-spread curve: maximum standardized entropy $\Phi_{\mu }$ that is feasible for a specified K-spread coefficient $K_{2}/\mu$ (Gini index). A particular state, defined as a point ($K_{2}/\mu ,\Phi_{\mu }$) is feasible only if it lies below this curve. The exact curve corresponds to the generalized half logistic (GHL) distribution, while different approximations (practically indistinguishable from the exact curve) are also plotted. Purple dashed lines show the boundaries between the partitioned areas.

**Figure 7 entropy-28-00035-f007:**
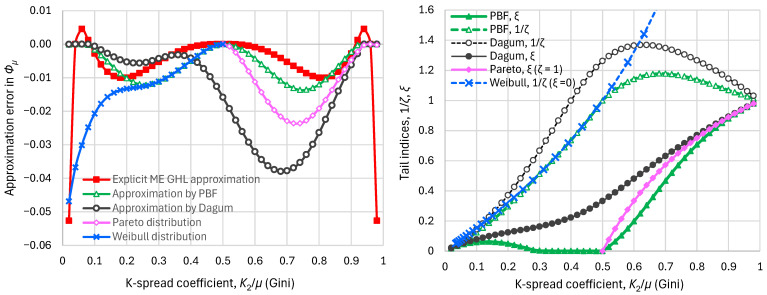
Approximations of the maximum entropy vs. K-spread curve: (**left**) approximation errors, defined as differences of the standardized entropy derived from approximations minus the exact values of the ME GHL distribution; (**right**) tail indices of the approximating distributions.

**Figure 8 entropy-28-00035-f008:**
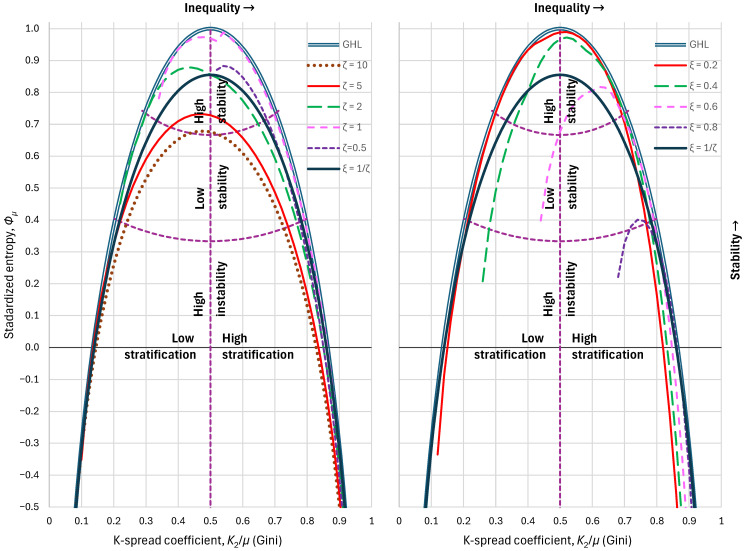
Maximum standardized entropy, as a function of the K-spread coefficient, attained by the PBF and Dagum distributions when (**left**) the lower tail index ζ is specified to the values shown in the legend and (**right**) the upper tail index ξ is specified to the values shown in the legend. Purple dashed lines show the boundaries between the partitioned areas.

**Figure 9 entropy-28-00035-f009:**
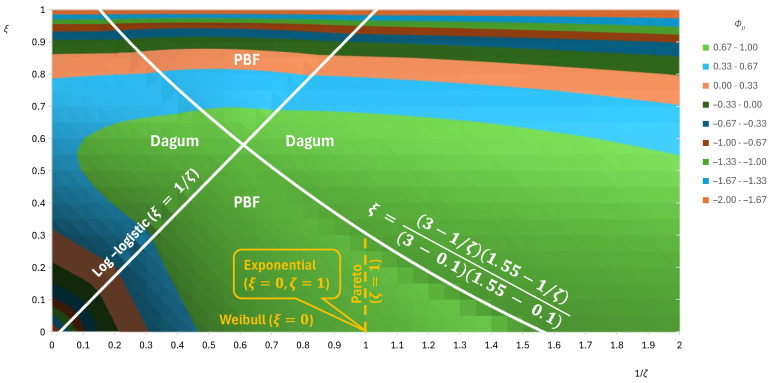
Contour plot of the attained maximum standardized entropy $\Phi_{\mu }$ by the PBF and the Dagum distributions as a function of the lower and upper tail index ζ,ξ, respectively. The two lines plotted in white separate the total area into four parts; in two, noted as “PBF”, the maximum is attained by the PBF distribution, while in the other two, noted as “Dagum”, the maximum is attained by the Dagum distribution. One of the two boundary lines depicts the log-logistic distribution, which is a special case of both the PBF and the Dagum distributions. The other boundary curve is derived after a systematic numerical investigation. The exponential, Pareto, and Weibull distributions, all of which are special cases of the PBF distribution, are also shown.

**Figure 10 entropy-28-00035-f010:**
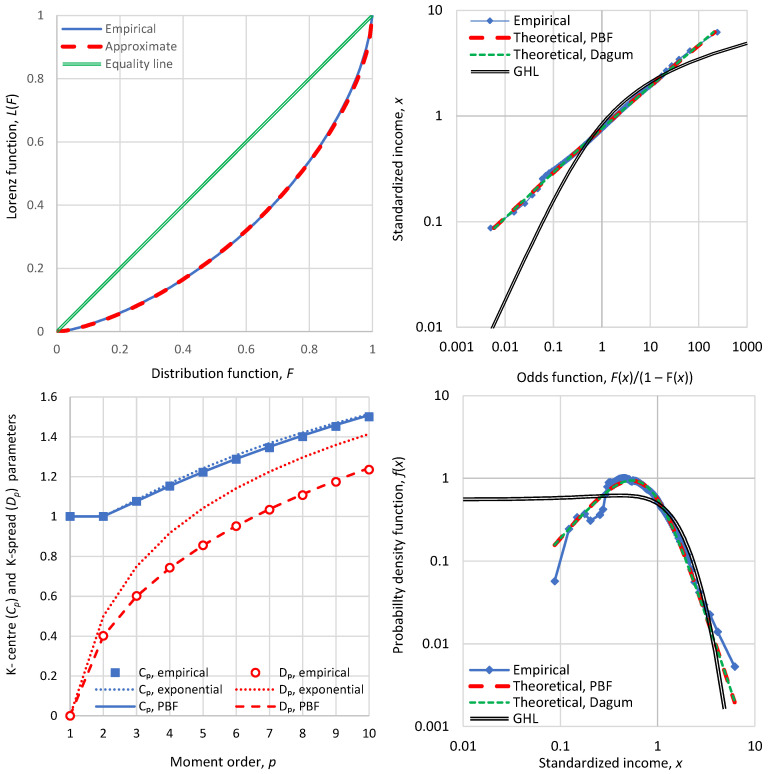
Detailed graphs of economic indicators in the USA in 2022 (**clockwise from upper left**): Lorenz curve; odds function; probability density function; and graph of K-centre and K-spread vs. K-moment order, where, for reference, the theoretical curves of the ME exponential distribution are also plotted in dotted lines.

**Figure 11 entropy-28-00035-f011:**
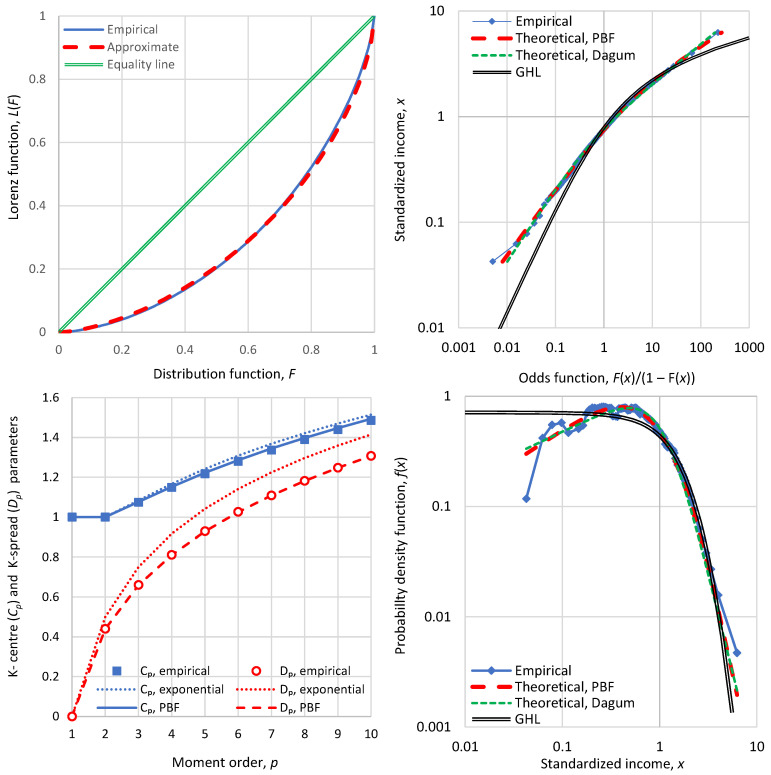
Detailed graphs of economic indicators in China in 2022 (**clockwise from upper left**): Lorenz curve; odds function; probability density function; and graph of K-centre and K-spread vs. K-moment order, where, for reference, the theoretical curves of the ME exponential distribution are also plotted in dotted lines.

**Figure 12 entropy-28-00035-f012:**
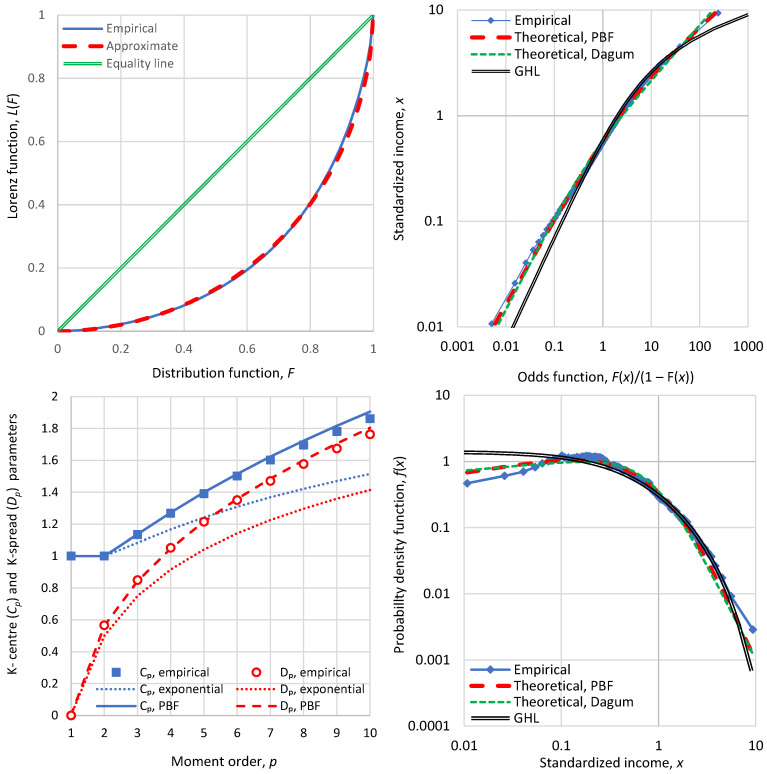
Detailed graphs of economic indicators of the World (more specifically, 69 countries from which data for 2022 are available, with a total population of 5.4 billion), (**clockwise from upper left**): Lorenz curve; odds function; probability density function; and graph of K-centre and K-spread vs. K-moment order, where, for reference, the theoretical curves of the ME exponential distribution are also plotted in dotted lines.

**Figure 13 entropy-28-00035-f013:**
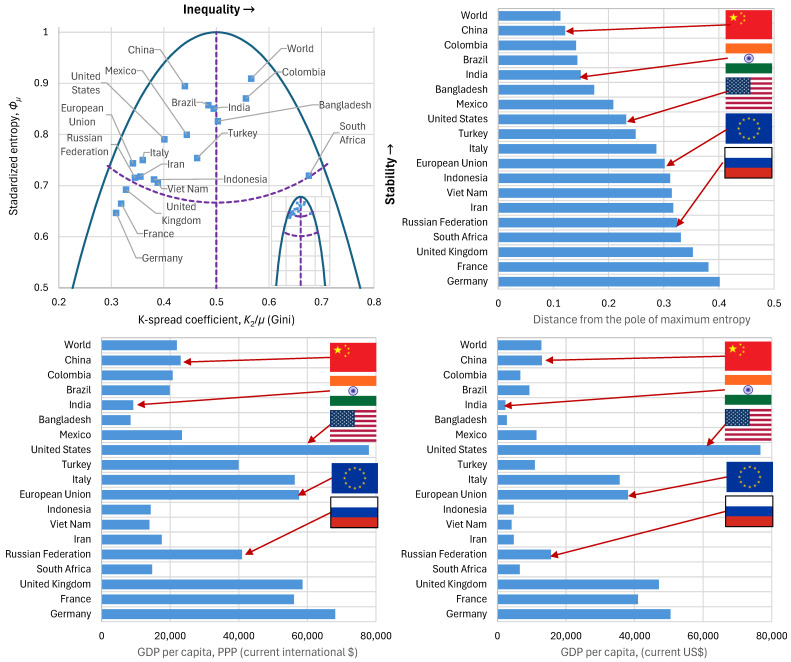
Characteristic graphs for the examined large population countries with data availability in 2022 (**clockwise from upper left**): standardized entropy vs. K-spread coefficient (Gini index), plotted alongside the maximum entropy vs. K-spread curve (purple dashed lines show the boundaries between the partitioned areas); distance from the pole of maximum entropy; GDP per capita [28]; GDP-PPP per capita, [29]. The inset in the upper left graph contains the entire range as in Figure 6. Five major geopolitical players are flagged. South Africa is included as one of the case studies (see below) even though the latest available data are for 2017.

**Figure 15 entropy-28-00035-f015:**
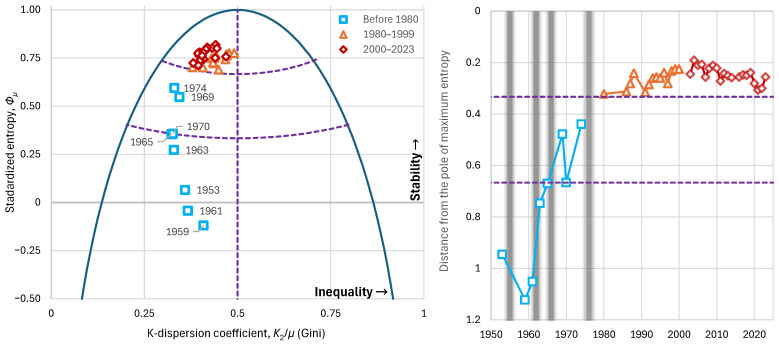
Characteristic graphs for the evolution of major economic indices in Argentina: (**left**): standardized entropy vs. K-spread coefficient (Gini index), plotted alongside the maximum entropy vs. K-spread curve; (**right**) distance from the pole of maximum entropy with the grey lines indicating the times of coups d’état. Purple dashed lines show the boundaries between the partitioned areas.

**Figure 16 entropy-28-00035-f016:**
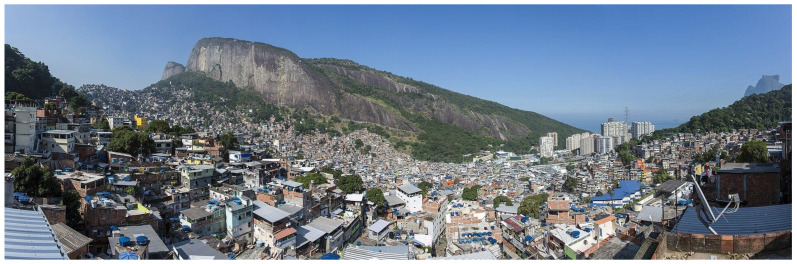
Panoramic view of Rio’s Rocinha favela, contrasted by high-rise buildings (condominiums) near the coast (of South Atlantic Ocean) in São Conrado [44].

**Figure 17 entropy-28-00035-f017:**
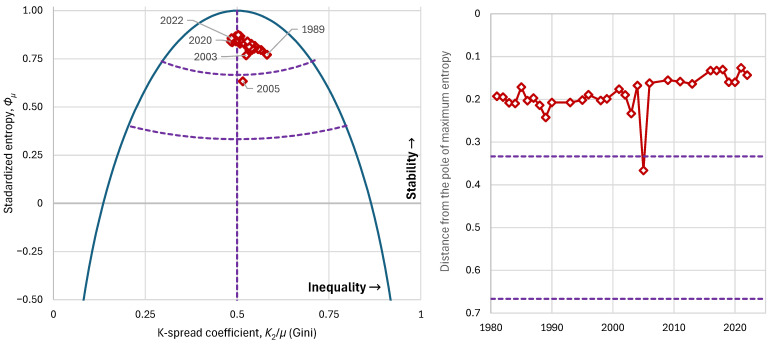
Characteristic graphs for the evolution of major economic indices in Brazil: (**left**): standardized entropy vs. K-spread coefficient (Gini index), plotted alongside the maximum entropy vs. K-spread curve; (**right**) distance from the pole of maximum entropy. Purple dashed lines show the boundaries between the partitioned areas.

**Figure 18 entropy-28-00035-f018:**
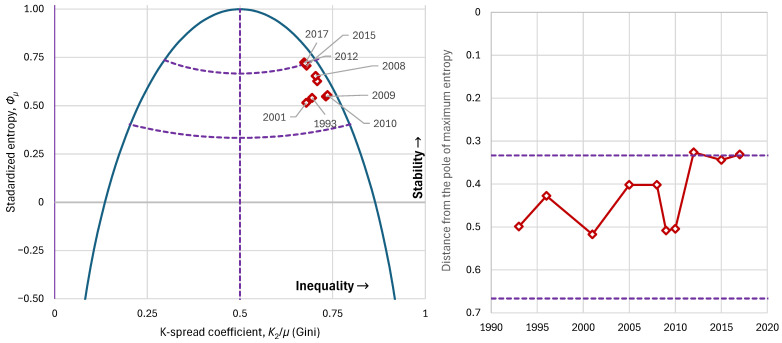
Characteristic graphs for the evolution of major economic indices in South Africa: (**left**): standardized entropy vs. K-spread coefficient (Gini index), plotted alongside the maximum entropy vs. K-spread curve; (**right**) distance from the pole of maximum entropy. Purple dashed lines show the boundaries between the partitioned areas.

**Table 1 entropy-28-00035-t001:** Distribution functions used in this study and their main characteristics *.

$\mathrm{Distribution},\;F\left(x\right)$	$\mathrm{Density},\;f\left(x\right)$	$\mathrm{K-Moments},\;K_{p}^{\prime }$ $\;\mathrm{and}/\mathrm{or}\;{\bar{K}}_{p}^{\prime }$	Mean, μ	$\mathrm{K-Spread}\;\mathrm{Coefficient},\;K_{2}/\mu$	$\mathrm{Standardized}\;\mathrm{Entropy},\;\Phi_{\mu }$
Exponential, $1-e^{-x/\lambda }$	$\frac{1}{\lambda }\bar{F}\left(x\right)$	$K_{p}^{\prime }=\lambda H_{p},\;{\bar{K}}_{p}^{\prime }=\frac{\lambda }{p}$	λ	$\frac{1}{2}$	1
Bounded exponential, $\frac{1-e^{-x/\lambda }}{1-e^{-\Omega /\lambda }}$	$\frac{1}{\lambda }\frac{1}{e^{x/\lambda }-1}F\left(x\right)$	$K_{p}^{\prime }=\Omega -\lambda {\left(1-e^{-\frac{\Omega }{\lambda }}\right)}^{-p}B_{1-e^{-\Omega /\lambda }}(p+1,0)$	$\lambda +\frac{\Omega }{1-e^{\Omega /\lambda }}$	$\frac{1}{2}\left(\frac{\Omega /\lambda }{\Omega /\lambda -e^{\Omega /\lambda }+1}+\coth\left(\frac{\Omega }{2\lambda }\right)\right)$	$1+\frac{\Omega /\lambda }{1-e^{\Omega /\lambda }}+\ln\left(\frac{2-2\cosh\left(\Omega /\lambda \right)}{\Omega /\lambda -e^{\Omega /\lambda }+1}\right)$
Logistic, $1-\frac{1}{1+e^{-\varsigma +x/\lambda }}$	$\frac{e^{-\varsigma +x/\lambda }}{\lambda }\bar{F}{\left(x\right)}^{2}$	$K_{p}^{\prime }=\lambda \left(H_{p-1}+\varsigma \right)$	λς	$\frac{1}{\varsigma }$	$2-\ln\left(\varsigma \right)$
GHL, $1-\frac{1}{1-e^{-\varsigma }+e^{-\varsigma +x/\lambda }}$	$\frac{\varsigma e^{-\varsigma +x/\lambda }}{\lambda }\bar{F}{\left(x\right)}^{2}$	$K_{p}^{\prime }=\lambda \left(H_{p}+\varsigma +\frac{B_{1-e^{\varsigma }}\left(p+1,0\right)}{{\left(1-e^{\varsigma }\right)}^{p}}\right)$	$\frac{\lambda \varsigma }{1-e^{-\varsigma }}$	$\frac{1}{\varsigma }\left(1-\frac{\varsigma }{e^{\varsigma }-1}\right)$	$2-\frac{\varsigma }{1-e^{-\varsigma }}-\ln\left(\frac{\varsigma }{e^{\varsigma }-1}\right)$
Pareto (ζ=1), ${1-\left(1+\xi \frac{x}{\lambda }\right)}^{-\frac{1}{\xi }}$	$\frac{1}{\lambda +\xi x}\bar{F}\left(x\right)$	$K_{p}^{\prime }=\frac{\lambda }{\xi }\left(pB\left(p,1-\xi \right)-1\right),\;\;\;{\bar{K}}_{p}^{\prime }=\frac{\lambda }{p-\xi }$	$\frac{\lambda }{1-\xi }$	$\frac{1}{2-\xi }$	1+ξ+ln(1−ξ)
Weibull (ξ=0), $1-\exp\left(-{\left(\frac{x}{\lambda }\right)}^{\zeta }\right)$	$\frac{\zeta }{\lambda }{\left(\frac{x}{\lambda }\right)}^{\zeta -1}\bar{F}\left(x\right)$	${\bar{K}}_{p}^{\prime }=\lambda p^{-1/\zeta }\Gamma \left(1+\frac{1}{\zeta }\right)$	$\frac{\lambda }{\zeta }\Gamma \left(\frac{1}{\zeta }\right)$	$1-2^{-1/\zeta }$	$1+\left(1-\frac{1}{\zeta }\right)\gamma -ln\Gamma \left(\frac{1}{\zeta }\right)$
PBF, $1-{\left(1+\zeta \xi {\left(\frac{x}{\lambda }\right)}^{\zeta }\right)}^{-\frac{1}{\zeta \xi }}$	$\frac{\zeta /x}{{\left(\frac{x}{\lambda }\right)}^{-\zeta }+\zeta \xi }\bar{F}\left(x\right)$	${\bar{K}}_{p}^{\prime }=\frac{\lambda \;p}{{\left(\zeta \xi \right)}^{1/\zeta }}B\left(1+\frac{1}{\zeta },\frac{p}{\zeta \xi }-\frac{1}{\zeta }\right)$	As ${\bar{K}}_{1}^{\prime }$	$1-\frac{B\left(\frac{1}{\zeta },\frac{2-\xi }{\zeta \xi }\right)}{B\left(\frac{1}{\zeta },\frac{1-\xi }{\zeta \xi }\right)}$	$1+\zeta \xi +\ln\left(\zeta \xi \right)+\left(1-\frac{1}{\zeta }\right)\left(\psi \left(\frac{1}{\zeta \xi }\right)+\gamma \right)-lnB\left(\frac{1}{\zeta },\frac{1-\xi }{\zeta \xi }\right)$
Dagum, ${\left(1+\frac{1}{\zeta \xi }{\left(\frac{x}{\lambda }\right)}^{-\frac{1}{\xi }}\right)}^{-\zeta \xi }$	$\frac{\zeta /x}{{\left(\frac{x}{\lambda }\right)}^{1/\xi }+\zeta \xi }F\left(x\right)$	$K_{p}^{\prime }=\lambda p{\left(\xi \zeta \right)}^{1-\xi }B\left(1-\xi ,p\zeta \xi +\xi \right)$	As $K_{1}^{\prime }$	$\frac{2B(1-\xi ,2\zeta \xi +\xi )}{B(1-\xi ,\zeta \xi +\xi )}-1$	$1+\frac{1}{\zeta \xi }-\ln\left(\zeta^{2}\xi \right)+\left(1+\xi \right)\left(\psi \left(\zeta \xi \right)+\gamma \right)-lnB\left(1-\xi ,\zeta \xi +\xi \right)$
Log-logistic, $1/\left(1+{\left(\frac{x}{\lambda }\right)}^{-\zeta }\right)$	$\frac{\zeta /x}{{\left(\frac{x}{\lambda }\right)}^{\zeta }+1}F\left(x\right)$	$K_{p}^{\prime }=\lambda p\;B\left(1-\frac{1}{\zeta },p+\frac{1}{\zeta }\right)$ ${\bar{K}}_{p}^{\prime }=\lambda p\;B\left(1+\frac{1}{\zeta },p-\frac{1}{\zeta }\right)$	$\frac{\pi \lambda }{\zeta \sin\frac{\pi }{\zeta }}$	$\frac{1}{\zeta }$	$\ln\left(\frac{1}{\pi }\sin\frac{\pi }{\zeta }\right)$

* Clarifications of symbols: γ=0.577216 is the Euler’s constant; $\Gamma \left(\cdot \right)$ and $B\left(\cdot ,\cdot \right)$ are the gamma and beta functions; $\psi \left(\cdot \right)$ is the digamma function; $H_{p}$ is the *p*th harmonic number; λ is a scale parameter; ξ and ζ are the upper and lower tail indices, respectively; ς is a shape parameter. The support of the bounded exponential distribution is (0,Ω). The support of the logistic distribution is (−∞,∞) and its upper and lower tail indices are $\xi =\xi^{\prime }=0$. For all other distributions, the support is (0,∞) and the upper and lower tail indices are ξ and ζ, respectively; when expressions do not include either of the two, their values are ξ=0, ζ=1. Note that when ξ=1/ζ the PBF and Dagum distributions yield the log-logistic.

**Table 2 entropy-28-00035-t002:** Characteristic indices for the economies of the major countries and composite entities in 2022.

Country, Year	$K_{2}/\mu$	$\Phi_{\mu }$	$d_{p}$	$D_{10}/D_{2}$	ξ	ζ
World	0.57	0.91	0.11	3.11	0.41	1.26
China	0.44	**0.89**	**0.12**	2.97	0.34	1.98
Colombia	0.56	0.87	0.14	3.31	0.53	1.67
Brazil	0.48	0.86	0.14	3.22	0.49	1.86
India	0.49	0.85	0.15	3.13	0.50	*1.02*
Bangladesh	0.50	0.83	0.17	3.44	0.69	2.59
Mexico	0.44	0.80	0.21	3.23	0.53	2.45
USA	0.40	0.79	0.23	3.08	0.41	2.31
Turkey	0.46	0.75	0.25	3.39	*0.72*	2.29
Italy	0.36	0.75	0.29	**2.94**	0.34	2.25
European Union	0.34	0.74	0.30	3.00	0.35	1.70
Indonesia	0.38	0.71	0.31	3.04	0.33	2.11
Viet Nam	0.39	0.71	0.31	3.14	0.41	2.65
Iran	0.36	0.72	0.32	3.03	0.41	2.24
Russian Federation	0.35	0.72	0.32	2.97	**0.31**	3.48
South Africa (2017)	0.68	0.72	0.33	*3.45*	0.55	2.24
United Kingdom	0.33	0.69	0.35	2.98	0.33	3.01
France	0.32	0.66	0.38	3.05	0.41	2.98
Germany	**0.31**	*0.65*	*0.40*	2.98	0.32	**3.51**
Bulgaria (1971)	0.23	−0.67	1.69	3.34	0.20	5.84

Note: All data are for the year 2022 except South Africa (2017). Bulgaria (1971), also shown in Figure 4, is included as an extreme case for comparison. The lowest or highest values among countries of the indices that favour equality or stability are highlighted in bold and those disfavouring them in italics, namely equality is manifested by low $K_{2}/\mu ,\;D_{10}/D_{2},\;\xi$ and high ζ, while stability is reflected in high $\Phi_{\mu }$ and low $d_{p}$.

## Data Availability

No new data were created; the data used are described in Section 2 and are publicly available in the given link.

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
