# Peer review of "Trade-Off Between Entropy and Gini Index in Income Distribution"

_entropy, 2025, doi:10.3390/e28010035_

Round 1

Reviewer 1 Report

Comments and Suggestions for Authors

Dear Professor Koutsoyiannis,

In my opinion, your manuscript is solid, coherent, mathematically rigorous, and presents a unified methodology  connecting entropy, K-moments, and inequality. The results are interesting and suitable for its publication in Entropy. 

I greatly enjoyed reading your article.  However, I have some minor comments and suggestions that I believe are pertinent. 

My recommendation is : Accept after minor revision

Comments for the authors

1) In Figure 1 (upper-right panel), the Lorenz curve (blue line) does not start at the origin. Please, correct it.

2) The analysis in the figures is based on visual comparisons rather than on a formal fitting procedure. It would be helpful if the authors could briefly justify this methodological choice and comment on why visual assessment is appropriate in this context.

3) The statements suggesting that “entropy declines can be linked to political ruptures” are intriguing but in my opinion not supported by systematic evidence. Please, either justify this connection more clearly or present it as a hypothesis rather than an established fact.

4) The introduction of an upper bound Ω is theoretically interesting, but real income distributions exhibit an unbounded Pareto type upper tail . It would be helpful if the authors clarified this issue.

5) Interpreting Φ_μ as an indicator of stability is appealing but not strictly derived from the mathematical results. Please, clarify this point.

With these minor revisions, the article is suitable for publication.

Reviewer 2 Report

Comments and Suggestions for Authors

The various images in this draft are rich. Otherwise, I suggest a major revision based on the following reasons. 1. I have concerns regarding the methodological rigor in interpreting the empirical results. Specifically, I believe the use of flexible models like the Pareto–Burr–Feller and Dagum distributions, although theoretically sound, may lead to overfitting, especially when dealing with limited data samples. In my view, the draft would benefit from a more detailed discussion on model validation techniques, such as cross-validation or out-of-sample testing, to ensure the robustness of the results. Furthermore, I think the interpretation of the tail indices (ξ, ζ) should be more cautious, as their estimation can be sensitive to outliers and sample size. This sensitivity could potentially lead to misleading conclusions about the distribution's tail behavior. 2. The draft's focus on a select group of countries (Argentina, Brazil, South Africa, Bulgaria, China, India, USA, Russia, EU) limits the generalizability of its findings. While these countries provide diverse economic and political contexts, the exclusion of other significant economies and regions may introduce bias and limit the applicability of the proposed framework. Expanding the analysis to include a broader range of countries, especially those with different levels of economic development and institutional frameworks, would enhance the external validity of the study. Furthermore, the draft could explore how cultural, social, and historical factors influence the relationship between entropy, inequality, and stability, providing a more nuanced understanding of the observed patterns. 3. While the draft offers a theoretically rich framework for understanding the tradeoff between entropy and the Gini index, it falls short in discussing the practical implications of its findings. The convex frontier between entropy and Gini reductions, although intriguing, lacks concrete policy recommendations or actionable insights for policymakers. The draft could benefit from a more detailed discussion on how the proposed indicators (K-spread, standardized entropy, tail indices) can be used to inform policy decisions, such as tax reforms, social welfare programs, or regulatory interventions. 4. Line 42, ‘spots’ the likeliest, is a bit colloquial. It is suggested to replace it with “identifies” or “determines”.

Round 2

Reviewer 2 Report

Comments and Suggestions for Authors

The authors replied to all my questions, so I suggest accepting the draft.